# Statistical Query Lower Bounds for List-Decodable Linear Regression

**Ilias Diakonikolas**
University of Wisconsin-Madison
ilias@cs.wisc.edu

**Daniel M. Kane**
University of California, San Diego
dakane@cs.ucsd.edu

**Ankit Pensia**
University of Wisconsin-Madison
ankitp@cs.wisc.edu

**Thanasis Pittas**
University of Wisconsin-Madison
pittas@wisc.edu

**Alistair Stewart**
Web 3 Foundation
stewart.al@gmail.com

## Abstract

We study the problem of list-decodable linear regression, where an adversary can corrupt a majority of the examples. Specifically, we are given a set $T$ of labeled examples $(x, y) \in \mathbb{R}^d \times \mathbb{R}$ and a parameter $0 < \alpha < 1/2$ such that an $\alpha$-fraction of the points in $T$ are i.i.d. samples from a linear regression model with Gaussian covariates, and the remaining $(1 - \alpha)$-fraction of the points are drawn from an arbitrary noise distribution. The goal is to output a small list of hypothesis vectors such that at least one of them is close to the target regression vector. Our main result is a Statistical Query (SQ) lower bound of $d^{\text{poly}(1/\alpha)}$ for this problem. Our SQ lower bound qualitatively matches the performance of previously developed algorithms, providing evidence that current upper bounds for this task are nearly best possible.

## 1 Introduction

### 1.1 Background and Motivation

Linear regression is one of the oldest and most fundamental statistical tasks with numerous applications in the sciences [RL87, Die01, McD09]. In the standard setup, the data are labeled examples $(x^{(i)}, y^{(i)})$, where the examples (covariates) $x^{(i)}$ are i.i.d. samples from a distribution $D_x$ on $\mathbb{R}^d$ and the labels $y^{(i)}$ are noisy evaluations of a linear function. More specifically, each label is of the form $y^{(i)} = \beta \cdot x^{(i)} + \eta^{(i)}$, where $\eta^{(i)}$ is the observation noise, for an unknown target regression vector $\beta \in \mathbb{R}^d$. The objective is to approximately recover the hidden regression vector. In this basic setting, linear regression is well-understood. For example, under Gaussian distribution, the least-squares estimator is known to be statistically and computationally efficient.

Unfortunately, classical efficient estimators inherently fail in the presence of even a very small fraction of adversarially corrupted data. In several applications of modern data analysis, including machine learning security [BNJT10, BNL12, SKL17, DKK+19] and exploratory data analysis, e.g., in biology [RPW+02, PLJD10, LAT+08], typical datasets contain arbitrary or adversarial outliers. Hence, it is important to understand the algorithmic possibilities and fundamental limits of learning and inference in such settings. Robust statistics focuses on designing estimators tolerant to a small

35th Conference on Neural Information Processing Systems (NeurIPS 2021).

amount of contamination, where the outliers are the *minority* of the dataset. Classical work in this field [HRRS86, HR09] developed robust estimators for various basic tasks, alas with exponential runtime. More recently, a line of work in computer science, starting with [DKK+16, LRV16], developed the first computationally efficient robust learning algorithms for various high-dimensional tasks. Subsequently, there has been significant progress in algorithmic robust statistics by several communities, see [DK19] for a survey on the topic.

In this paper, we study high-dimensional robust linear regression in the presence of a *majority* of adversarial outliers. As we explain below, in several applications, asking for a minority of outliers is too strong of an assumption. It is thus natural to ask what notion of learning can capture the regime when the clean data points (inliers) constitute the *minority* of the dataset. While outputting a *single* accurate hypothesis in this regime is information-theoretically impossible, one may be able to compute a *small list* of hypotheses with the guarantee that *at least one of them* is accurate. This relaxed notion is known as *list-decodable learning* [BBV08, CSV17], formally defined below.

**Definition 1.1.** (List-Decodable Learning) Given a parameter $0 < \alpha < 1/2$ and a distribution family $\mathcal{D}$ on $\mathbb{R}^d$, the algorithm specifies $n \in \mathbb{Z}_+$ and observes $n$ i.i.d. samples from a distribution $E = \alpha D + (1-\alpha)N$, where $D$ is an unknown distribution in $\mathcal{D}$ and $N$ is arbitrary. We say $D$ is the distribution of inliers, $N$ is the distribution of outliers, and $E$ is an $(1-\alpha)$-corrupted version of $D$. Given sample access to an $(1-\alpha)$-corrupted version of $D$, the goal is to output a "small" list of hypotheses $\mathcal{L}$ at least one of which is (with high probability) close to the target parameter of $D$.

We note that a list of size $O(1/\alpha)$ typically suffices; an algorithm with a $\text{poly}(1/\alpha)$ sized list, or even a worse function of $1/\alpha$ (but independent of the dimension $d$) is also considered acceptable.

Natural applications of list-decodable learning include crowdsourcing, where a majority of participants could be unreliable [SVC16, MV18], and semi-random community detection in stochastic block models [CSV17]. List-decoding is also useful in the context of semi-verified learning [CSV17, MV18], where a learner can audit a very small amount of trusted data. If the trusted dataset is too small to directly learn from, using a list-decodable learning procedure, one can pinpoint a candidate hypothesis consistent with the verified data. Importantly, list-decodable learning generalizes the task of learning mixture models, see, e.g., [DeV89, JJ94, ZJD16, LL18, KC20, CLS20, DK20] for the case of linear regression studied here. Roughly speaking, by running a list-decodable estimation procedure with the parameter $\alpha$ equal to the smallest mixing weight, each true cluster of points is an equally valid ground-truth distribution, so the output list must contain candidate parameters close to each of the true parameters.

In list-decodable linear regression (the focus of this paper), $D$ is a distribution on pairs $(X, y)$, where $X$ is a standard Gaussian on $\mathbb{R}^d$, $y$ is approximately a linear function of $x$, and the algorithm is asked to approximate the hidden regressor. The following definition specifies the distribution family $\mathcal{D}$ of the inliers for the case of linear regression with Gaussian covariates.

**Definition 1.2.** (Gaussian Linear Regression) Fix $\sigma > 0$. For $\beta \in \mathbb{R}^d$, let $D_\beta$ be the distribution over $(X, y)$, $X \in \mathbb{R}^d$, $y \in \mathbb{R}$, such that $X \sim \mathcal{N}(0, I_d)$ and $y = \beta^T X + \eta$, where $\eta \sim \mathcal{N}(0, \sigma^2)$ independently of $X$. We define $\mathcal{D}$ to be the set $\{D_\beta : \beta \in S'\}$ for some set $S' \subseteq \mathbb{R}^d$.

Recent algorithmic progress [KKK19, RY20a] has been made on this problem using the sum-of-squares (SoS) hierarchy. The guarantees in [KKK19, RY20a] are very far from the information-theoretic limit in terms of sample complexity. In particular, they require $d^{\text{poly}(1/\alpha)}$ samples and time to obtain non-trivial error guarantees (see Table 1): [KKK19] obtains an error guarantee of $O(\sigma/\alpha)$ with a list of size $O(1/\alpha)$, whereas [RY20a] obtains an error guarantee of $O(\sigma/\alpha^{3/2})$ with a list of size $(1/\alpha)^{O(\log(1/\alpha))}$.

On the other hand, as shown in this paper (see Theorem 1.4), $\text{poly}(d/\alpha)$ samples information-theoretically suffice to obtain near-optimal error guarantees. This raises the following natural question:

> *What is the complexity of list-decodable linear regression?*
> *Are there efficient algorithms with significantly better sample-time tradeoffs?*

We study the above question in a natural and well-studied restricted model of computation, known as the Statistical Query (SQ) model [Kea98]. As the main result of this paper, we prove strong SQ lower bounds for this problem. Via a recently established equivalence [BBH+20], our SQ lower bound also

Table 1: The table summarizes the sample complexity, running time, and list size of the known list-decodable linear regression algorithms in order to obtain a $1/4$-additive approximation to the hidden regression vector $\beta$ in the setting of Theorem 1.5, i.e., when $\|\beta\|_2 \leq 1$ and $\sigma$ is sufficiently small as a function of $\alpha$: [KKK19] requires $\sigma = O(\alpha)$ and [RY20a] requires $\sigma = O(\alpha^{3/2})$.

| Algorithmic Result | Sample Size | Running Time | List size |
|---|---|---|---|
| Karmalkar-Klivans-Kothari [KKK19] | $(d/\alpha)^{O(1/\alpha^4)}$ | $(d/\alpha)^{O(1/\alpha^8)}$ | $O(1/\alpha)$ |
| Raghavendra and Yau [RY20a] | $d^{O(1/\alpha^4)}$ | $d^{O(1/\alpha^8)}(1/\alpha)^{\log(1/\alpha)}$ | $(1/\alpha)^{O(\log(1/\alpha))}$ |

implies low-degree testing lower bounds for this task. Our lower bounds can be viewed as evidence that current upper bounds for this problem may be qualitatively best possible.

Before we state our contributions in detail, we give some background on SQ algorithms. SQ algorithms are a broad class of algorithms that are only allowed to query expectations of bounded functions of the distribution rather than directly access samples. Formally, an SQ algorithm has access to the following oracle.

**Definition 1.3** (STAT Oracle). Let $D$ be a distribution on $\mathbb{R}^d$. A statistical query is a bounded function $q : \mathbb{R}^d \to [-1, 1]$. For $\tau > 0$, the $\text{STAT}(\tau)$ oracle responds to the query $q$ with a value $v$ such that $|v - \mathbf{E}_{X \sim D}[q(X)]| \leq \tau$. We call $\tau$ the tolerance of the statistical query.

The SQ model was introduced by Kearns [Kea98] in the context of supervised learning as a natural restriction of the PAC model [Val84]. Subsequently, the SQ model has been extensively studied in a plethora of contexts (see, e.g., [Fel16] and references therein). The class of SQ algorithms is rather broad and captures a range of known supervised learning algorithms. More broadly, several known algorithmic techniques in machine learning are known to be implementable using SQs. These include spectral techniques, moment and tensor methods, local search (e.g., Expectation Maximization), and many others (see, e.g., [FGR$^+$17, FGV17]).

## 1.2 Our Results

We start by showing that $\text{poly}(d/\alpha)$ samples are sufficient to obtain a near-optimal error estimator, albeit with a computationally inefficient algorithm.

**Theorem 1.4** (Information-Theoretic Bound). *There is a (computationally inefficient) list-decoding algorithm for Gaussian linear regression that uses $O(d/\alpha^3)$ samples, returns a list of $O(1/\alpha)$ many hypothesis vectors, and has $\ell_2$-error guarantee of $O((\sigma/\alpha)\sqrt{\log(1/\alpha)})$. Moreover, if the dimension $d$ is sufficiently large, any list-decoding algorithm that outputs a list of size $\text{poly}(1/\alpha)$ must have $\ell_2$-error at least $\Omega((\sigma/\alpha)/\sqrt{\log(1/\alpha)})$.*

Due to space limitations, the proof of Theorem 1.4 is deferred to the supplementary material (see Theorems D.1 and D.4). We note that the (computationally inefficient) estimator achieving the upper bound in Theorem 1.4 is implicit in [KKK19]. See Appendix D.1 for more details.

Our main result is a strong SQ lower bound for the list-decodable Gaussian linear regression problem. We establish the following theorem (see Theorem 2.1 for a more detailed formal statement).

**Theorem 1.5** (SQ Lower Bound). *Assume that the dimension $d \in \mathbb{Z}_+$ is sufficiently large and consider the problem of list-decodable linear regression, where the fraction of inliers is $\alpha \in (0, 1/2)$, the regression vector $\beta \in \mathbb{R}^d$ has norm $\|\beta\|_2 \leq 1$, and the additive noise has standard deviation $\sigma \leq \alpha$. Then any SQ algorithm that returns a list $\mathcal{L}$ of candidate vectors containing a $\widehat{\beta}$ such that $\|\widehat{\beta} - \beta\|_2 \leq 1/4$ does one of the following: (i) it uses at least one query with tolerance at most $d^{-\Omega(1/\sqrt{a})}/\sigma$, (ii) it makes $2^{d^{\Omega(1)}}$ queries, or (iii) it returns a list of size $|\mathcal{L}| = 2^{d^{\Omega(1)}}$.*

Informally speaking, Theorem 1.5 shows that no SQ algorithm can approximate $\beta$ to constant accuracy with a sub-exponential in $d^{\Omega(1)}$ size list and sub-exponential in $d^{\Omega(1)}$ many queries, unless using queries of very small tolerance – that would require at least $\sigma d^{\Omega(1/\sqrt{\alpha})}$ samples to simulate. For $\sigma$ not too small, e.g., $\sigma = \text{poly}(\alpha)$, in view of Theorem 1.4, this result can be viewed as an information-computation tradeoff for the problem, within the class of SQ algorithms.

A conceptual implication of Theorem 1.5 is that list-decodable linear regression is harder (within the class of SQ algorithms) than the related problem of learning mixtures of linear regressions (MLR). Recent work [DK20] gave an algorithm (easily implementable in SQ) for learning MLR with $k$ equal weight separated components (under Gaussian covariates) with sample complexity and running time $k^{\mathrm{polylog}(k)}$, i.e., *quasi-polynomial* in $k$. Recalling that one can reduce $k$-MLR (with well-separated components) to list-decodable linear regression for $\alpha = 1/k$, Theorem 1.5 implies that the aforementioned algorithmic result cannot be obtained via such a reduction.

**Remark 1.6.** We note that our lower bounds rule out efficient algorithms in the SQ model, which is a broad class of algorithms. The two existing algorithms [RY20a, KKK19] for the present problem are based on the sum-of-squares (SoS) hierarchy. In general, SQ lower bounds do not imply lower bounds against all SoS algorithms, and SoS lower bounds do not imply SQ lower bounds. At the same time, lower bounds against low-degree tests [HS17, HKP+17, Hop18, KWB19] have become the standard heuristic for SoS lower bounds, and we establish hardness against low-degree tests as well. In particular, recent work [BBH+20] established that (under certain assumptions) an SQ lower bound also implies a qualitatively similar lower bound in the low-degree model. By leveraging this connection, we deduce a similar lower bound in the latter model as well (see Appendix F).

## 1.3 Overview of Techniques

In this section, we provide a detailed overview of our SQ lower bound construction. We recall that there exists a general methodology for establishing SQ lower bounds via an appropriate complexity measure, known as SQ dimension. Several related notions of SQ dimension exist in the literature, see, e.g., [BFJ+94, FGR+17, Fel17]. Here we focus on the framework introduced in [FGR+17] for search problems over distributions, which is more natural in our setting. A lower bound on the SQ dimension of a search problem provides an unconditional lower bound on the SQ complexity of the problem. Roughly speaking, for a notion of correlation between distributions in our family $\mathcal{D}$ (Definition 1.8), establishing an SQ lower bound amounts to constructing a large cardinality sub-family $\mathcal{D}' \subseteq \mathcal{D}$ such that every pair of distributions in $\mathcal{D}'$ are nearly uncorrelated with respect to a given reference distribution $R$ (see Definition 1.10 and Lemma 1.11).

A general framework for constructing SQ-hard families of distributions was introduced in [DKS17], which showed the following: Let the reference distribution $R$ be $\mathcal{N}(0, I)$ and $A$ be a univariate distribution whose low-degree moments match those of the standard Gaussian (and which satisfies an additional mild technical condition). Let $P_{A,v}$ be the distribution that is a copy of $A$ in the $v$-direction and standard Gaussian in the orthogonal complement (Definition 1.12). Then the distribution family $\{P_{A,v}\}_{v \in S}$, where $S$ is a set of nearly orthogonal unit vectors, satisfies the pairwise nearly uncorrelated property (Lemma 1.13), and is therefore SQ-hard to learn.

Unfortunately, the [DKS17] framework does not suffice in the supervised setting of the current paper for the following reason: The joint distribution over labeled examples $(X, y)$ in our setting does not possess the symmetry properties required for moment-matching with the reference $R = \mathcal{N}(0, I)$ to be possible. Specifically, the behavior of $y$ will necessarily be somewhat different than the behavior of $X$. To circumvent this issue, we leverage an idea from [DKS19]. The high-level idea is to construct distributions $E_v$ on $(X, y)$ such that for any fixed value $y_0$ of $y$, the conditional distribution of $X \mid y = y_0$ under $E_v$ is of the form $P_{A,v}$ described above, where $A$ is replaced with some $A_{y_0}$.

We further explain this modified construction. Note that $E_v$ should be of the form $\alpha D_v + (1-\alpha)N_v$, where $D_v$ is the inlier distribution (corresponding to the clean samples from the linear regression model) and $N_v$ is the outlier (noise) distribution. To understand what properties our distribution should satisfy, we start by looking at the inlier distribution $D$. By definition, for $(X, y) \sim D$, we have that $y = \beta^T X + \eta$, where $X \sim N(0, I)$ and $\eta \sim N(0, \sigma^2)$ is independent of $X$. A good place to start here is to understand the distribution of $X$ conditioned on $y = y_0$, for some $y_0$, under $D$. It is not hard to show (Fact 2.3) that this conditional distribution is already of the desired form $P_{A,\beta}$: it is a product of a $(d-1)$-dimensional standard Gaussian in directions orthogonal to $\beta$, while in the $\beta$-direction it is a much narrower Gaussian with mean proportional to $y_0$. To establish our SQ-hardness result, we would like to mix this conditional distribution with a carefully selected outlier distribution $N \mid y = y_0$, such that the resulting mixture $E \mid y = y_0$ matches many of its low-degree moments with the standard Gaussian in the $\beta$-direction, while being standard Gaussian in the orthogonal directions. In the setting of minority of outliers, [DKS19] was able to provide an explicit formula for $N$ and match *three* moments to show an SQ lower bound of $\Omega(d^2)$. The main

technical difficulty in our paper is that, in order to prove the desired SQ lower bound of $\Omega(d^{\mathrm{poly}(1/\alpha)})$, we need to match $\mathrm{poly}(1/\alpha)$ many moments. We explain how to achieve this below.

Here we take a different approach and establish the existence of the desired outlier distribution $N|y = y_0$ in a non-constructive manner. We note that our problem is an instance of the moment-matching problem, where given a sequence of real numbers, the goal is to decide whether a distribution exists having that sequence as its low-degree moments. At a high-level, we leverage classical results that tackle this general question by formulating a linear program (LP) and using LP-duality to derive necessary and sufficient feasibility conditions (see [KS53] and Theorem 3.1). This moment-matching via LP duality approach is fairly general, but stumbles upon two technical obstacles in our setting.

The first technical issue is that our final distributions $E_v$ on $(X, y)$ need to have bounded $\chi^2$-divergence with respect to the reference distribution, since the pairwise correlations scale with this quantity (see Lemma 1.13). To guarantee this, we can ensure that the outlier distribution in the $\beta$-direction is in fact equal to the convolution of a distribution with bounded support with a narrow Gaussian: (i) The contraction property of this convolution operator means that it can only reduce the $\chi^2$-divergence, and (ii) the bounded support can be used in combination with tail-bounds on Hermite polynomials (Lemma 3.6) to bound from above the contribution to the $\chi^2$-divergence of each Hermite coefficient of our distribution (Lemma 2.6). These additional constraints necessitate a modification to the moment-matching problem, but it can still be readily analyzed (Theorem 2.5).

The second and more complicated issue involves the fraction of outliers, i.e., the parameter "$1-\alpha$". Unfortunately, it is easy to see that the fraction of outliers necessary to make the conditional distributions match the desired number of moments must necessarily go to 1 as $|y|$ goes to infinity: As $|y|$ gets bigger, the conditional distribution of inliers moves further away from $\mathcal{N}(0, I)$ (Fact 2.3) and thus needs to be mixed more heavily with outliers to be corrected. This is a significant problem, since by definition we can only afford to use a $(1-\alpha)$-fraction of outliers overall. To handle this issue, we consider a reference distribution $R$ on $(X, y)$ that has much heavier tails in $y$ than the distribution of inliers has. This essentially means that as $|y|$ gets large, the conditional probability that a sample is an outlier gets larger and larger. This is balanced by having slightly lower fraction of outliers for smaller values of $|y|$, in order to ensure that the total fraction of outliers is still at most $1-\alpha$. To address this issue, we leverage the fact that the probability that a clean sample has large value of $|y|$ is very small. Consequently, we can afford to make the error rates for such $y$ quite large without increasing the overall probability of error by very much.

## 1.4 Preliminaries

**Notation** We use $\mathbb{N}$ to denote natural numbers and $\mathbb{Z}_+$ to denote positive integers. For $n \in \mathbb{Z}_+$ we denote $[n] \overset{\text{def}}{=} \{1, \ldots, n\}$ and use $\mathcal{S}^{d-1}$ for the $d$-dimensional unit sphere. We denote by $\mathbf{1}(\mathcal{E})$ the indicator function of the event $\mathcal{E}$. We use $I_d$ to denote the $d \times d$ identity matrix. For a random variable $X$, we use $\mathbf{E}[X]$ for its expectation. For $m \in \mathbb{Z}_+$, the $m$-th moment of $X$ is defined as $\mathbf{E}[X^m]$. We use $\mathcal{N}(\mu, \Sigma)$ to denote the Gaussian distribution with mean $\mu$ and covariance matrix $\Sigma$. We let $\phi$ denote the pdf of the one-dimensional standard Gaussian. When $D$ is a distribution, we use $X \sim D$ to denote that the random variable $X$ is distributed according to $D$. For a vector $x \in \mathbb{R}^d$, we let $\|x\|_2$ denote its $\ell_2$-norm. For $y \in \mathbb{R}$, we denote by $\delta_y$ the Dirac delta distribution at $y$, i.e., the distribution that assigns probability mass 1 to the single point $y \in \mathbb{R}$ and zero elsewhere. When there is no confusion, we will use the same letters for distributions and their probability density functions.

**Ornstein-Uhlenbeck Operator** For a $\rho > 0$, we define the *Gaussian noise* (or *Ornstein-Uhlenbeck*) operator $U_\rho$ as the operator that maps a distribution $F$ on $\mathbb{R}$ to the distribution of the random variable $\rho X + \sqrt{1 - \rho^2} Z$, where $X \sim F$ and $Z \sim \mathcal{N}(0, 1)$ independently of $X$.

**Background on the SQ Model** We provide the basic definitions and facts that we use.

**Definition 1.7** (Search problems over distributions). Let $\mathcal{D}$ be a set of distributions over $\mathbb{R}^d$, $\mathcal{F}$ be a set called solutions, and $\mathcal{Z} : \mathcal{D} \to 2^{\mathcal{F}}$ be a map that assigns sets of solutions to distributions of $\mathcal{D}$. The *distributional search problem* $\mathcal{Z}$ over $\mathcal{D}$ and $\mathcal{F}$ is to find a valid solution $f \in \mathcal{Z}(D)$ given statistical query oracle access to an unknown $D \in \mathcal{D}$.

The hardness of these problems is conveniently captured by the SQ dimension. For this, we first need to define the notion of correlation between distributions.

**Definition 1.8** (Pairwise Correlation). The pairwise correlation of two distributions with probability density functions $D_1, D_2 : \mathbb{R}^d \to \mathbb{R}_+$ with respect to a reference distribution with density $R : \mathbb{R}^d \to \mathbb{R}_+$, where the support of $R$ contains the supports of $D_1$ and $D_2$, is defined as $\chi_R(D_1, D_2) \overset{\text{def}}{=} \int_{\mathbb{R}^d} D_1(x) D_2(x) / R(x) \, dx - 1$. When $D_1 = D_2$, the pairwise correlation becomes the same as the $\chi^2$-divergence between $D_1$ and $R$, i.e., $\chi^2(D_1, R) \overset{\text{def}}{=} \int_{\mathbb{R}^d} D_1^2(x) / R(x) dx - 1$.

**Definition 1.9.** For $\gamma, \beta > 0$, the set of distributions $\mathcal{D} = \{D_1, \ldots, D_m\}$ is called $(\gamma, \beta)$-correlated relative to the distribution $R$ if $|\chi_R(D_i, D_j)| \leq \gamma$, if $i \neq j$, and $|\chi_R(D_i, D_j)| \leq \beta$ otherwise.

The statistical dimension of a search problem is based on the largest set of $(\gamma, \beta)$-correlated distributions assigned to each solution.

**Definition 1.10** (Statistical Dimension). For $\gamma, \beta > 0$, a search problem $\mathcal{Z}$ over a set of solutions $\mathcal{F}$ and a class $\mathcal{D}$ of distributions over $X$, we define the *statistical dimension* of $\mathcal{Z}$, denoted by $\mathrm{SD}(\mathcal{Z}, \gamma, \beta)$, to be the largest integer $m$ such that there exists a reference distribution $R$ over $X$ and a finite set of distributions $\mathcal{D}_R \subseteq \mathcal{D}$ such that for any solution $f \in \mathcal{F}$, the set $\mathcal{D}_f = \mathcal{D}_R \setminus \mathcal{Z}^{-1}(f)$ is $(\gamma, \beta)$-correlated relative to $R$ and $|\mathcal{D}_f| \geq m$.

**Lemma 1.11** (Corollary 3.12 in [FGR$^+$17]). *Let $\mathcal{Z}$ be a search problem over a set of solutions $\mathcal{F}$ and a class of distributions $\mathcal{D}$ over $\mathbb{R}^d$. For $\gamma, \beta > 0$, let $s = \mathrm{SD}(\mathcal{Z}, \gamma, \beta)$ be the statistical dimension of the problem. For any $\gamma' > 0$, any SQ algorithm for $\mathcal{Z}$ requires either $s\gamma'/(\beta - \gamma)$ queries or at least one query to $\mathrm{STAT}(\sqrt{\gamma + \gamma'})$ oracle.*

We continue by recalling the machinery from [DKS17] that will be used for our construction.

**Definition 1.12** (High-Dimensional Hidden Direction Distribution). For a unit vector $v \in \mathbb{R}^d$ and a distribution $A$ on the real line with probability density function $A(x)$, define $P_{A,v}$ to be a distribution over $\mathbb{R}^d$, where $P_{A,v}$ is the product distribution whose orthogonal projection onto the direction of $v$ is $A$, and onto the subspace perpendicular to $v$ is the standard $(d-1)$-dimensional normal distribution. That is, $P_{A,v}(x) := A(v^T x) \phi_{\perp v}(x)$, where $\phi_{\perp v}(x) = \exp\left(-\|x - (v^T x)v\|_2^2 / 2\right) / (2\pi)^{(d-1)/2}$.

The distributions $\{P_{A,v}\}$ defined above are shown to be nearly uncorrelated as long as the directions where $A$ is embedded are pairwise nearly orthogonal.

**Lemma 1.13** (Lemma 3.4 in [DKS17]). *Let $m \in \mathbb{Z}_+$. Let $A$ be a distribution over $\mathbb{R}$ that agrees with the first $m$ moments of $\mathcal{N}(0, 1)$. For any $v$, let $P_{A,v}$ denote the distribution from Definition 1.12. For all $v, u \in \mathbb{R}^d$, we have that $\chi_{\mathcal{N}(0, I_d)}(P_{A,v}, P_{A,u}) \leq |u^T v|^{m+1} \chi^2(A, \mathcal{N}(0, 1))$.*

The following result shows that there are exponentially many nearly-orthogonal unit vectors.

**Lemma 1.14** (see, e.g., Lemma 3.7 of [DKS17]). *For any $0 < c < 1/2$, there is a set $S$, of at least $2^{\Omega(d^c)}$ unit vectors in $\mathbb{R}^d$, such that for each pair of distinct $v, v' \in S$, it holds $|v^T v'| \leq O(d^{c-1/2})$.*

## 1.5 Prior and Related Work

Early work in robust statistics, starting with the pioneering works of Huber and Tukey [Hub64, Tuk75], pinned down the sample complexity of high-dimensional robust estimation with a minority of outliers. In contrast, until relatively recently, even the most basic computational questions in this field were poorly understood. Two concurrent works [DKK$^+$16, LRV16] gave the first provably robust and efficiently computable estimators for robust mean and covariance estimation. Since the dissemination of these works, there has been a flurry of activity on algorithmic robust estimation in a variety of high-dimensional settings; see [DK19] for a recent survey on the topic. Notably, the robust estimators developed in [DKK$^+$16] are scalable in practice and yield a number of applications in exploratory data analysis [DKK$^+$17] and adversarial machine learning [TLM18, DKK$^+$19]

The list-decodable learning setting studied in this paper was first considered in [CSV17] with a focus on mean estimation. [CSV17] gave a polynomial-time algorithm with near-optimal statistical guarantees for list-decodable mean estimation under a bounded covariance assumption on the clean. Subsequent work has led to significantly faster algorithms for the bounded covariance setting [DKK20a, CMY20, DKK$^+$20b, DKK$^+$21] and polynomial-time algorithms with improved error guarantees under stronger distributional assumptions [DKS18, KSS18]. More recently, a line of work developed list-decodable learners for more challenging tasks, including linear regression [KKK19, RY20a] and subspace recovery [RY20b, BK21].

## 2 Main Result: Proof of Theorem 1.5

In this section, we present the main result of this paper: SQ hardness of list-decodable linear regression (Definitions 1.1 and 1.2). We consider the setting when $\beta$ has norm less than 1, i.e., $\beta = \rho v$ for $v \in \mathcal{S}^{d-1}$ and $\rho \in (0,1)$.[1] Note that the marginal distribution of the labels is $\mathcal{N}(0, \sigma_y^2)$, where $\sigma_y^2 = \rho^2 + \sigma^2$. We ensure that the labels $y$ have unit variance by using $\sigma^2 = 1 - \rho^2$. Specifically, the choice of parameters will be such that obtaining a $\rho/2$-additive approximation of the regressor $\beta$ is possible information-theoretically with $\text{poly}(d/\alpha)$ samples (cf. Appendix D.1), but the complexity of any SQ algorithm for the task must necessarily be at least $d^{\text{poly}(1/\alpha)}/\sigma$. We show the following more detailed statement of Theorem 1.5.

**Theorem 2.1** (SQ Lower Bound). *Let $c \in (0, 1/2)$, $d \in \mathbb{Z}_+$ with $d = 2^{\Omega(1/(1/2-c))}$, $\alpha \in (0, 1/2)$, $\rho \in (0,1)$, $\sigma^2 = 1 - \rho^2$, and $m \in \mathbb{Z}_+$ with $m \leq c_1/\sqrt{\alpha}$ for some sufficiently small constant $c_1 > 0$. Any list-decoding algorithm that, given statistical query access to a $(1-\alpha)$-corrupted version of the distribution described by the model of Definition 1.2 with $\beta = \rho v$ for $v \in \mathcal{S}^{d-1}$, returns a list $\mathcal{L}$ of hypotheses vectors that contains a $\widehat{\beta}$ such that $\|\widehat{\beta} - \beta\|_2 \leq \rho/2$, does one of the following: (i) it uses at least one query to $\text{STAT}\left(\Omega(d)^{-(2m+1)(1/4-c/2)}e^{O(m)}/\sqrt{1-\rho^2}\right)$, (ii) it makes $2^{\Omega(d^c)}d^{-(2m+1)(1/2-c)}$ many queries, or (iii) it returns a list $\mathcal{L}$ of size $2^{\Omega(d^c)}$.*

In the rest of this section, we will explain the hard-to-learn construction for our SQ lower bound, i.e., a set of distributions with large statistical dimension. The proof would then follow from Lemma 1.11. We begin by describing additional notation that we will use.

**Notation**: As $\beta = \rho v$ for a fixed $\rho$, we will slightly abuse notation by using $D_v(x, y)$ to denote the joint distribution of the inliers and we use $E_v(x, y)$ to denote the $(1-\alpha)$-corrupted version of $D_v(x, y)$. To avoid using multiple subscripts, we use $D_v(x|y)$ to denote the conditional distribution of $X|y$ according to the distribution $D_v$ and similarly for the other distributions. In addition, we use $D_v(y)$ to denote the marginal distribution of $y$ under $D_v$ and similarly for other distributions.

Following the general construction of [DKS17], we will specify a *reference* joint distribution $R(x, y)$ where $X$ and $y$ are independent, and $X \sim \mathcal{N}(0, I_d)$. We will find a marginal distribution $R(y)$ and a way to add the outliers so that the following hold for each $E_v$ (where $m = \Theta(1/\sqrt{\alpha})$):

(I) $E_v$ is indeed a valid distribution of $(X, y)$ in our corruption model (i.e., can be written as a mixture $\alpha D_v(x, y) + (1-\alpha)N_v(x, y)$ for some noise distribution $N_v$). Moreover, the marginal of $E_v$ on the labels, $E_v(y)$, coincides with $R(y)$.

(II) For every $y \in \mathbb{R}$, the conditional distribution $E_v(x|y)$ is of the form $P_{A_y, v}$ of Definition 1.12, with $A_y$ being a distribution that matches the first $2m$ moments with $\mathcal{N}(0, 1)$.[2]

(III) For $A_y$ defined above, $\mathbf{E}_{y \sim R(y)}[\chi^2(A_y, \mathcal{N}(0, 1))]$ is bounded.

We first briefly explain why a construction satisfying the above properties suffices to prove our main theorem (postponing a formal proof for the end of this section). We start by noting the following decomposition (proved in Appendix B).

**Lemma 2.2.** *For $u, v \in \mathcal{S}^{d-1}$, if $E_u$ and $E_v$ have the same marginals $R(y)$ on the labels, they satisfy $\chi_{R(x,y)}(E_v(x, y), E_u(x, y)) = \mathbf{E}_{y \sim R(y)}\left[\chi_{\mathcal{N}(0, I_d)}(E_v(x|y), E_u(x|y))\right].$*

Using the decomposition in Lemma 2.2 for $E_u$ and $E_v$ satisfying Property (II), Lemma 1.13 implies that $|\chi_{R(x,y)}(E_v(x, y), E_u(x, y))| \leq |u^T v|^{2m+1} \mathbf{E}_{y \sim R(y)}[\chi^2(A_y, N(0, 1))]$. Letting $\mathcal{D} = \{E_v : v \in S\}$, where $S$ is the set of nearly uncorrelated unit vectors from Lemma 1.14, we get that $\mathcal{D}$ is $(\gamma, b)$-correlated relative to $R$, for $b = \mathbf{E}_{y \sim R(y)}[\chi^2(A_y, \mathcal{N}(0, 1))]$ and $\gamma \leq d^{-\Omega(m)}b$. As $|S| = 2^{\Omega(d^c)}$, $b$ is bounded, and the list size is much smaller than $|S|$, we can show that the statistical dimension of the list-decodable linear regression is large.

Thus, in the rest of the section we focus on showing that such a construction exists. We first note that according to our linear model of Definition 1.2, the conditional distribution of $X$ given $y$ for the inliers is Gaussian with unit variance in all but one direction (see Appendix B for a proof).

---

[1]This is a standard assumption and considered by existing works [KKK19, RY20a] (cf. Remark B.4).

[2]We use even number of moments for simplicity. The analysis would slightly differ for odd number.

**Fact 2.3.** *Fix $\rho > 0$, $v \in \mathcal{S}^{d-1}$, and consider the regression model of Definition 1.2 with $\beta = \rho v$. Then the conditional distribution $X|y$ of the inliers is $\mathcal{N}(y\rho v, I_d - \rho^2 vv^T)$, i.e., independent standard Gaussian in all directions perpendicular to $v$ and $\mathcal{N}(\rho y, 1 - \rho^2)$ in the direction of $v$.*

Since Fact 2.3 states that $D_v(x|y)$ is already of the desired form (standard normal in all directions perpendicular to $v$ and $\mathcal{N}(y\rho, 1 - \rho^2)$ in the direction of $v$), the problem becomes one-dimensional. More specifically, for every $y \in \mathbb{R}$, we need to find a one-dimensional distribution $Q_y$ and appropriate values $\alpha_y \in [0, 1]$ such that the mixture $A_y = \alpha_y \mathcal{N}(y\rho, 1 - \rho^2) + (1 - \alpha_y)Q_y$ matches the first $2m$ moments with $\mathcal{N}(0, 1)$. Then, multiplying by $\phi_{\perp v}$ (which denotes the contribution of the space orthogonal to $v$ to the density of standard Gaussian, as defined in Definition 1.12) yields the $d$-dimensional mixture distribution $\alpha_y D_v(x|y) + (1 - \alpha_y)Q_y(v^T x)\phi_{\perp v}(x)$. We show that an appropriate selection of $\alpha_y$ can ensure that this is a valid distribution for our contamination model.

**Lemma 2.4.** *Let $R$ be a distribution on pairs $(x, y) \in \mathbb{R}^{d+1}$ such that $\alpha_y := \alpha D_v(y)/R(y) \in [0, 1]$ for all $y \in \mathbb{R}$. Suppose that for every $y \in \mathbb{R}$ there exists a univariate distribution $Q_y$ such that $A_y := \alpha_y \mathcal{N}(y\rho, 1 - \rho^2) + (1 - \alpha_y)Q_y$ matches the first $2m$ moments with $\mathcal{N}(0, 1)$. If the distribution of the outliers is $N_v(x, y) = ((1 - \alpha_y)/(1 - \alpha))Q_y(v^T x)\phi_{\perp v}(x)R(y)$, Properties (I) and (II) hold.*

The proof of Lemma 2.4 is included in Appendix B. We will choose the reference distribution $R(x, y)$ to have $X \sim \mathcal{N}(0, I_d)$ and $y \sim \mathcal{N}(0, 1/\alpha)$ independently, which makes the corresponding value of $\alpha_y$ to be $\alpha_y = \alpha D_v(y)/R(y) = \sqrt{\alpha}\exp(-y^2(1 - \alpha)/2)$. This satisfies the condition in Lemma 2.4 that $\alpha_y \in [0, 1]$. Our choice of $R(y) \sim \mathcal{N}(0, 1/\alpha)$ is informed by Properties (II) and (III), and will be used later on in the proofs of Theorem 2.5 and Lemma 2.6 (also see the last paragraph of Section 1.3 for more intuition). Going back to our goal, i.e., making $A_y = \alpha_y \mathcal{N}(y\rho, 1 - \rho^2) + (1 - \alpha_y)Q_y$ match moments with $\mathcal{N}(0, 1)$, we will argue that it suffices to only look for $Q_y$ of the specific form $U_\rho F_y$, where $U_\rho$ is the Ornstein-Uhlenbeck operator. This suffices because $U_\rho \delta_y = \mathcal{N}(y\rho, 1 - \rho^2)$ and the operator $U_\rho$ preserves the moments of a distribution if they match with $\mathcal{N}(0, 1)$ (see Lemma 2.6 (i) below). Letting $A_y = U_\rho(\alpha_y \delta_y + (1 - \alpha_y)F_y)$, the new goal is to show that the argument of $U_\rho$ matches moments with $\mathcal{N}(0, 1)$. We show the following structural result:

**Theorem 2.5.** *Let $y \in \mathbb{R}$, $B \in \mathbb{R}$, $\alpha \in (0, 1/2)$, and define $\alpha_y := \sqrt{\alpha}\exp(-y^2(1 - \alpha)/2)$. For any $m \in \mathbb{Z}_+$ such that $m \leq C_1/\sqrt{\alpha}$ and $B \geq C_2\sqrt{m}$, with $C_1 > 0$ being a sufficiently small constant and $C_2$ being a sufficiently large constant, there exists a distribution $F_y$ that satisfies the following:*

1. *The mixture distribution $\alpha_y \delta_y + (1 - \alpha_y)F_y$ matches the first $2m$ moments with $\mathcal{N}(0, 1)$.*

2. *$F_y$ is a discrete distribution supported on at most $2m + 1$ points, all of which lie in $[-B, B]$.*

The proof of Theorem 2.5 is the bulk of the technical work of this paper and is deferred to Section 3. As mentioned before, applying $U_\rho$ preserves the required moment-matching property. More crucially, it allows us to bound the $\chi^2$-divergence: the following result bounds $\chi^2(A_y, \mathcal{N}(0, 1))$ using contraction properties of $U_\rho$, tail bounds of Hermite polynomials, and the discreteness of $F_y$.

**Lemma 2.6.** *In the setting of Theorem 2.5, let $\rho > 0$ and $Q_y = U_\rho F_y$. Then the following holds for the mixture $A_y = \alpha_y \mathcal{N}(y\rho, 1 - \rho^2) + (1 - \alpha_y)Q_y$: (i) $A_y$ matches the first $2m$ moments with $\mathcal{N}(0, 1)$, and (ii) $\chi^2(A_y, \mathcal{N}(0, 1)) \leq \alpha O(e^{y^2(\alpha - 1/2)})/(1 - \rho^2) + O(e^{B^2/2})/(1 - \rho^2)$.*

We prove Lemma 2.6 in Appendix B. We are now ready to sketch the proof of Theorem 2.1 (see Appendix B for the detailed proof).

*Proof Sketch of Theorem 2.1.* Consider the search problem $\mathcal{Z}$, where $\mathcal{D}$ is the set of all distributions $E_v$ satisfying properties (I),(II), and (III) (let $\beta(v) = \rho v$ be the corresponding regressors). For each $E_v$, the corresponding solution set is defined to consist of all lists $\mathcal{L}$ of size $\ell$ having one element that is $(\rho/2)$-close to $\beta(v)$. Let the subset $\mathcal{D}_R = \{E_v\}_{v \in S}$, for $S$ being the set of nearly orthogonal vectors of Lemma 1.14. Since $|u^T v| \leq O(d^{c-1/2})$ for any distinct $u, v \in S$ and $d = 2^{\Omega(1/(1/2-c))}$, for any vector $w$, at most one element of $S$ can be $(\rho/2)$-close to $w$. Thus, for any list $\mathcal{L}$ of size $\ell = |S|/2$, $|\mathcal{D}_R \setminus \mathcal{Z}^{-1}(\mathcal{L})| \geq |S| - \ell \geq 2^{\Omega(d^c)}$. Using Lemmas 2.2 and 1.13 along with the $\chi^2$-bound of Lemma 2.6, we get that $\mathcal{D}_R$ is $(\gamma, b)$-correlated with respect to $R$, for $b := e^{O(m)}/(1 - \rho^2)$ and $\gamma := \Omega(d)^{-(2m+1)(1/2-c)}b$. An application of Lemma 1.11 completes the proof. $\qquad\square$

# 3 Duality for Moment Matching: Proof of Theorem 2.5

We now prove the existence of a bounded distribution $F_y$ such that the mixture $\alpha_y \delta_y + (1-\alpha_y)F_y$ matches the first $2m$ moments with $\mathcal{N}(0,1)$. The proof follows a non-constructive argument based on the duality between the space of moments and the space of non-negative polynomials.

Let $B > 0$ and $m \in \mathbb{Z}_+$. Let $\mathcal{P}(m)$ denote the class of all polynomials $p : \mathbb{R} \to \mathbb{R}$ with degree at most $m$. Let $\mathcal{P}^{\geq 0}(2m, B)$ be the class of polynomials that can be represented in either the form $p(t) = (\sum_{i=0}^{m} a_i t^i)^2$ or the form $p(t) = (B^2 - t^2)(\sum_{i=0}^{m-1} b_i t^i)^2$. The intuition for $\mathcal{P}^{\geq 0}(2m, B)$ is that every polynomial of degree at most $2m$ that is non-negative in $[-B, B]$ can be written as a finite sum of polynomials from $\mathcal{P}^{\geq 0}(2m, B)$. By slightly abusing notation, for a polynomial $p(t) = \sum_{i=0}^{m} p_i t^i$, we also use $p$ to denote the vector in $\mathbb{R}^{m+1}$ consisting of the coefficients $(p_0, \ldots, p_m)$. The following classical result characterizes when a vector is realizable as the moment sequence of a distribution with support in $[-B, B]$ (for simplicity, we focus on matching an even number of moments in the rest of this section).

**Theorem 3.1** (Theorem 16.1 of [KS53]). *Let $B > 0$, $k \in \mathbb{Z}_+$, and $x = (x_0, x_1, \ldots, x_{2k}) \in \mathbb{R}^{2k+1}$ with $x_0 = 1$. There exists a distribution with support in $[-B, B]$ having as its first $2k$ moments the sequence $(x_1, \ldots, x_{2k})$ if and only if for all $p \in \mathcal{P}^{\geq 0}(2k, B)$ it holds that $\sum_{i=0}^{2k} x_i p_i \geq 0$.*

As we require the distribution to be discrete, we prove the following result using Theorem 3.1:

**Proposition 3.2.** *Fix $y \in \mathbb{R}$, $\alpha_y \in (0,1)$, $B > 0$, and $m \in \mathbb{Z}_+$. There exists a discrete distribution $F_y$ supported on at most $2m + 1$ points in $[-B, B]$ such that $\alpha_y \delta_y + (1 - \alpha_y)F_y$ matches the first $2m$ moments with $\mathcal{N}(0,1)$ if and only if $\mathbf{E}_{X \sim \mathcal{N}(0,1)}[p(X)] \geq \alpha_y p(y)$ for all $p \in \mathcal{P}^{\geq 0}(2m, B)$.*

The proof of Proposition 3.2 is deferred to Appendix C.1. To prove Theorem 2.5, we need to establish the condition of Proposition 3.2. To this end, we first need the following two technical lemmas, whose proofs are sketched towards the end of this section (for detailed proofs see Sections C.2 and C.3).

**Lemma 3.3.** *Let $m \in \mathbb{Z}_+$. If $B \geq C\sqrt{m}$ for some sufficiently large constant $C > 0$, then for every $q \in \mathcal{P}(m)$, it holds that $B^2 \mathbf{E}_{X \sim \mathcal{N}(0,1)}[q^2(X)] \geq 2 \mathbf{E}_{X \sim \mathcal{N}(0,1)}[X^2 q^2(X)]$.*

**Lemma 3.4.** *Let $y \in \mathbb{R}$, $\alpha \in (0, 1/2)$, $m \in \mathbb{Z}_+$, and $\alpha_y = \sqrt{\alpha} \exp(-y^2(1 - \alpha)/2)$. Suppose $m \leq C/\sqrt{\alpha}$ for some sufficiently small constant $C > 0$. Then for all $r \in \mathcal{P}(m), r \not\equiv 0$: $r^2(y)/(\mathbf{E}_{X \sim \mathcal{N}(0,1)}[r^2(X)]) \leq 1/(2\alpha_y)$.*

*Proof of Theorem 2.5.* By Proposition 3.2, it remains to show that if $B \geq C_2\sqrt{m}$, then the condition $\mathbf{E}_{X \sim \mathcal{N}(0,1)}[p(X)] \geq \alpha_y p(y)$ holds for all $p \in \mathcal{P}^{\geq 0}(2m, B)$. Thus, it suffices to ensure that the following two inequalities hold for $X \sim \mathcal{N}(0,1)$:

$$\sup_{r \in \mathcal{P}(m), r \not\equiv 0} \frac{r^2(y)}{\mathbf{E}[r^2(X)]} \leq \frac{1}{\alpha_y} \quad \text{and} \quad \sup_{q \in \mathcal{P}(m-1), q \not\equiv 0} \frac{(B^2 - y^2)q^2(y)}{\mathbf{E}[(B^2 - X^2)q^2(X)]} \leq \frac{1}{\alpha_y}, \qquad (1)$$

where we use Lemma 3.3 to show that $\mathbf{E}[(B^2 - X^2)q^2(X)] > 0$ for all non-zero polynomials $q \in \mathcal{P}(m - 1)$. The first expression can be bounded using Lemma 3.4 when $m \leq C_1/\sqrt{\alpha}$. We now focus on the second expression. By Lemma 3.3, $\mathbf{E}_{X \sim \mathcal{N}(0,1)}[(B^2 - X^2)q^2(X)] \geq 0.5 \mathbf{E}_{X \sim \mathcal{N}(0,1)}[B^2 q^2(X)]$. Therefore, we have that

$$\sup_{q \in \mathcal{P}(m-1), q \not\equiv 0} \frac{(B^2 - y^2)q^2(y)}{\mathbf{E}_{X \sim \mathcal{N}(0,1)}[(B^2 - X^2)q^2(X)]} \leq \sup_{q \in \mathcal{P}(m-1), q \not\equiv 0} \frac{B^2 q^2(y)}{\mathbf{E}_{X \sim \mathcal{N}(0,1)}[(B^2 - X^2)q^2(X)]}$$

$$\leq \sup_{q \in \mathcal{P}(m-1), q \not\equiv 0} \frac{B^2 q^2(y)}{\mathbf{E}_{X \sim \mathcal{N}(0,1)}[0.5B^2 q^2(X)]} = 2 \sup_{q \in \mathcal{P}(m-1), q \not\equiv 0} \frac{q^2(y)}{\mathbf{E}_{X \sim \mathcal{N}(0,1)}[q^2(X)]},$$

where the first inequality uses that the denominator is positive and $y^2 q^2(y) \geq 0$ and the second inequality uses that $\mathbf{E}_{X \sim \mathcal{N}(0,1)}[(B^2 - X^2)q^2(X)] \geq 0.5 \mathbf{E}_{X \sim \mathcal{N}(0,1)}[B^2 q^2(X)]$. The expression above is of the same form as the first expression in Equation (1), and thus is also bounded above by $1/\alpha_y$ when $m \leq C_1/\sqrt{\alpha}$ using Lemma 3.4. This completes the proof of Theorem 2.5. $\qquad \square$

**Proof sketch of Lemma 3.3:** The proof of Lemma 3.3 is a relatively straightforward application of Hölder's inequality and the Gaussian Hypercontractivity Theorem (stated below). For $p \in (0, \infty)$, we define the $L^p$-norm of a random variable $X$ to be $\|X\|_{L^p} := (\mathbf{E}[|X|^p])^{1/p}$.

**Fact 3.5** (Gaussian Hypercontractivity [Bog98, Nel73]). *Let $X \sim \mathcal{N}(0, 1)$. If $p \in \mathcal{P}(d)$ and $t \geq 2$, then $\|p(X)\|_{L^t} \leq (t-1)^{d/2} \|p(X)\|_{L^2}$.*

**Proof sketch of Lemma 3.4:** The proof is based on Hermite Analysis (see Appendix A for more details). The *normalized probabilist's Hermite polynomials*, $\{h_i, i \in [m]\}$ form a basis of $\mathcal{P}(m)$ and satisfy the property $\mathbf{E}_{X \sim \mathcal{N}(0,1)}[h_i(X)h_j(X)] = \mathbf{1}(i = j)$. Since $r$ is a polynomial of degree at most $m$, we can represent $r(x) = \sum_{i=1}^{m} a_i h_i(x)$ for some $a_i \in \mathbb{R}$. Using orthonormality of $h_i$ under the Gaussian measure, we get that $\mathbf{E}_{X \sim \mathcal{N}(0,1)}[r^2(X)] = \sum_{i=1}^{m} a_i^2$. By a standard optimization argument, we get that the supremum of $r^2(y)/\mathbf{E}[r^2(X)]$ is exactly $\sum_{i=1}^{m} h_i^2(y)$. It remains to show that for every $y \in \mathbb{R}$, $\sum_{i=1}^{m} \alpha_y h_i^2(y) \leq 1/2$. As $m \leq C/\sqrt{\alpha}$ for a small enough constant $C$, it suffices to show that for every $i \in [m]$, $\alpha_y h_i^2(y) \leq O(\sqrt{\alpha})$. As $\alpha_y := \sqrt{\alpha} \exp(-y^2(1-\alpha)/2)$, the following tail bound on the Hermite polynomials can be used:

**Lemma 3.6.** *Let $h_i$ be the $i$-th normalized probabilist's Hermite polynomial. Then $\max_{x \in \mathbb{R}} h_k^2(x)e^{-x^2/2} = O(k^{-1/6})$.*

We break our analysis in two cases:

**Case 1:** $|y| \leq 1/\sqrt{\alpha}$. Since $\alpha^2 y \leq 1$, Lemma 3.6 implies that for every $|y| \leq 1/\sqrt{\alpha}$, $\alpha_y h_i^2(y) = \sqrt{\alpha} \exp(1)h_i^2(y) \exp(-y^2/2) = O(\sqrt{\alpha})$.

**Case 2:** $|y| > 1/\sqrt{\alpha}$. In this case, we use rather crude bounds. A direct calculation shows that $|h_i(x)| \leq i^i(1 + |x|)^i$. Since $\alpha \in (0, 1/2)$, we get that $\alpha_y h_i^2(y) \leq \sqrt{\alpha} \exp(-y^2/4 + 2i \log(2i|y|))$. It remains to show that $\exp(-y^2/4 + 2i \log(2i|y|)) = O(1)$ under given conditions on $i$ and $y$. We have that $\exp(-y^2/4 + 2i \log(2i|y|)) = O(1)$ whenever $|y| = \Omega(\sqrt{i \log i})$. Since $|y| \geq 1/\sqrt{\alpha}$, the former condition is satisfied whenever $i = O(1/\sqrt{\alpha})$. This completes the proof sketch. $\qquad \square$

### Acknowledgments

Ilias Diakonikolas is supported by NSF Award CCF-1652862 (CAREER), a Sloan Research Fellowship, and a DARPA Learning with Less Labels (LwLL) grant. Daniel M. Kane is supported by NSF Award CCF-1553288 (CAREER) and a Sloan Research Fellowship. Ankit Pensia is supported by NSF Award DMS-1749857. Thanasis Pittas is supported in part by NSF Award DMS-2023239 (TRIPODS).

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
