# Appendix

## A  Omitted Background: Basics on Hermite Analysis

Hermite polynomials form a complete orthogonal basis of the vector space $L^2(\mathbb{R}, \mathcal{N}(0,1))$ of all functions $f : \mathbb{R} \to \mathbb{R}$ such that $\mathbf{E}_{X \sim \mathcal{N}(0,1)}[f^2(X)] < \infty$. There are two commonly used types of Hermite polynomials. The *physicist's* Hermite polynomials, denoted by $H_k$ for $k \in \mathbb{N}$ satisfy the following orthogonality property with respect to the weight function $e^{-x^2}$: for all $k, m \in \mathbb{N}$, $\int_{\mathbb{R}} H_k(x) H_m(x) e^{-x^2} \mathrm{dx} = \sqrt{\pi} 2^k k! \mathbf{1}(k = m)$. The *probabilist's* Hermite polynomials $H_{e_k}$ for $k \in \mathbb{N}$ satisfy $\int_{\mathbb{R}} H_{e_k}(x) H_{e_m}(x) e^{-x^2/2} \mathrm{dx} = k! \sqrt{2\pi} \mathbf{1}(k = m)$ and are related to the physicist's polynomials through $H_{e_k}(x) = 2^{-k/2} H_k(x/\sqrt{2})$. We will mostly use the *normalized probabilist's* Hermite polynomials, $h_k(x) = H_{e_k}(x)/\sqrt{k!}$, $k \in \mathbb{N}$ for which $\int_{\mathbb{R}} h_k(x) h_m(x) e^{-x^2/2} \mathrm{dx} = \sqrt{2\pi} \mathbf{1}(k = m)$. These polynomials are the ones obtained by Gram-Schmidt orthonormalization of the basis $\{1, x, x^2, \ldots\}$ with respect to the inner product $\langle f, g \rangle_{\mathcal{N}(0,1)} = \mathbf{E}_{X \sim \mathcal{N}(0,1)}[f(X)g(X)]$. Every function $f \in L^2(\mathbb{R}, \mathcal{N}(0,1))$ can be uniquely written as $f(x) = \sum_{i \in \mathbb{N}} a_i h_i(x)$ and we have $\lim_{n \to n} \mathbf{E}_{x \sim \mathcal{N}(0,1)}[(f(x) - \sum_{i=0}^{n} a_i h_i(x))^2] = 0$ (see, e.g., [AAR99]).

We now state a well-known property of *Ornstein–Uhlenbeck* operator that we use in our proofs, which is the fact that $U_\rho$ operates diagonally with respect to Hermite polynomials.

**Fact A.1** (see, e.g., [O'D14]). *For any Hermite polynomial $h_i$, any distribution $F$ on $\mathbb{R}$, and $\rho \in (0,1)$, it holds that $\mathbf{E}_{X \sim U_\rho F}[h_i(X)] = \rho^i \mathbf{E}_{X \sim F}[h_i(X)]$.*

## B  Omitted Proofs from Section 2

In this section we restate and prove the following results.

**Lemma 2.2.** *For $u, v \in \mathcal{S}^{d-1}$, if $E_u$ and $E_v$ have the same marginals $R(y)$ on the labels, they satisfy*
$$\chi_{R(x,y)}(E_v(x,y), E_u(x,y)) = \mathbf{E}_{y \sim R(y)} \left[ \chi_{\mathcal{N}(0,I_d)} (E_v(x|y), E_u(x|y)) \right].$$

*Proof.* Let $\phi$ denote the density of $\mathcal{N}(0,1)$. Using the fact that $E_v$ and $E_u$ have the same marginal $R(y)$ we have that

$$
\begin{aligned}
\chi_{R(x,y)}(E_v(x,y), E_u(x,y)) + 1 &= \int_{\mathbb{R}} \int_{\mathbb{R}^d} \frac{E_v(x,y) E_u(x,y)}{\phi(x) R(y)} \mathrm{dxdy} \\
&= \int_{\mathbb{R}} \int_{\mathbb{R}^d} \frac{E_v(x|y) E_u(x|y)}{\phi(x)} R(y) \mathrm{dxdy} \\
&= \int_{\mathbb{R}} \left( 1 + \chi_{\mathcal{N}(0,I_d)}(E_v(x|y) E_u(x|y)) \right) R(y) \mathrm{dy} \\
&= 1 + \mathop{\mathbf{E}}_{y \sim R(y)} \left[ \chi_{\mathcal{N}(0,I_d)} (E_v(x|y), E_u(x|y)) \right].
\end{aligned}
$$

$\square$

**Fact 2.3.** *Fix $\rho > 0$, $v \in \mathcal{S}^{d-1}$, and consider the regression model of Definition 1.2 with $\beta = \rho v$. Then the conditional distribution $X|y$ of the inliers is $\mathcal{N}(y \rho v, I_d - \rho^2 v v^T)$, i.e., independent standard Gaussian in all directions perpendicular to $v$ and $\mathcal{N}(\rho y, 1 - \rho^2)$ in the direction of $v$.*

*Proof.* This is due to the following fact for the conditional distribution of the Gaussian distribution.

**Fact B.1.** *If $\begin{bmatrix} y_1 \\ y_2 \end{bmatrix} \sim \mathcal{N}\left( \begin{bmatrix} \mu_1 \\ \mu_2 \end{bmatrix}, \begin{bmatrix} \Sigma_{11} & \Sigma_{12} \\ \Sigma_{21} & \Sigma_{22} \end{bmatrix} \right)$, then $y_1|y_2 \sim \mathcal{N}(\bar{\mu}, \bar{\Sigma})$, with $\bar{\mu} = \mu_1 + \Sigma_{12} \Sigma_{22}^{-1}(y_2 - \mu_2)$ and $\Sigma_{11} - \Sigma_{12} \Sigma_{22}^{-1} \Sigma_{21}$.*

We apply this fact for the pair $(X, y)$ by setting $y_1 = X, y_2 = y, \mu_1 = \mu_2 = 0$ and $\Sigma_{11} = I_d, \Sigma_{12} = \beta, \Sigma_{21} = \beta^T, \Sigma_{22} = \sigma^2 + \|\beta\|_2^2$. $\square$

**Lemma 2.4.** *Let $R$ be a distribution on pairs $(x, y) \in \mathbb{R}^{d+1}$ such that $\alpha_y := \alpha D_v(y)/R(y) \in [0, 1]$ for all $y \in \mathbb{R}$. Suppose that for every $y \in \mathbb{R}$ there exists a univariate distribution $Q_y$ such that $A_y := \alpha_y \mathcal{N}(y\rho, 1 - \rho^2) + (1-\alpha_y)Q_y$ matches the first $2m$ moments with $\mathcal{N}(0, 1)$. If the distribution of the outliers is $N_v(x, y) = ((1-\alpha_y)/(1-\alpha))Q_y(v^T x)\phi_{\perp v}(x)R(y)$, Properties (I) and (II) hold.*

*Proof.* First note that the noise distribution $N_v$ is indeed a valid distribution since it is non-negative everywhere because of the assumption $\alpha_y \in [0, 1]$ and it integrates to one:

$$\frac{1}{1-\alpha} \int_{\mathbb{R}} \int_{\mathbb{R}^d} (1 - \alpha_y)Q_y(v^T x)\phi_{\perp v}(x)R(y)\mathrm{dxdy}$$

$$= \frac{1}{1-\alpha} \left( \int_{\mathbb{R}} \int_{\mathbb{R}^d} R(y)Q_y(v^T x)\phi_{\perp v}(x)\mathrm{dxdy} - \alpha \int_{\mathbb{R}} \int_{\mathbb{R}^d} D_v(y)Q_y(v^T x)\phi_{\perp v}(x)\mathrm{dxdy} \right) = 1 \ .$$

The joint distribution $E_v(x, y)$ can be written as

$$E_v(x, y) = \alpha D_v(x, y) + (1 - \alpha)N_v(x, y)$$

$$= \alpha D_v(x, y) + (1 - \alpha)\frac{1 - \alpha_y}{1 - \alpha}Q_y(v^T x)\phi_{\perp v}(x)R(y)$$

$$= \left( \alpha_y D_v(x|y) + (1 - \alpha_y)Q_y(v^T x)\phi_{\perp v}(x) \right) R(y) \ .$$

This means that the marginal of $y$ under $E_v$ is $R(y)$, which establishes Property (I), and the conditional distribution of $X|y$ under $E_v$ is $E_y(x|y) = \alpha_y D_v(x|y) + (1 - \alpha_y)Q_y(v^T x)\phi_{\perp v}(x)$.

The moment matching part of Property (II) holds by assumption. For the other part of Property (II), we note that $E_v(x|y)$ is standard Gaussian in directions perpendicular to $v$ because of Fact 2.3 and the form of the term $Q_y(v^T x)\phi_{\perp v}(x)$ that corresponds to the outliers. □

**Lemma 2.6.** *In the setting of Theorem 2.5, let $\rho > 0$ and $Q_y = U_\rho F_y$. Then the following holds for the mixture $A_y = \alpha_y \mathcal{N}(y\rho, 1 - \rho^2) + (1-\alpha_y)Q_y$: (i) $A_y$ matches the first $2m$ moments with $\mathcal{N}(0, 1)$, and (ii) $\chi^2(A_y, \mathcal{N}(0, 1)) \leq \alpha O(e^{y^2(\alpha-1/2)})/(1 - \rho^2) + O(e^{B^2/2})/(1 - \rho^2)$.*

*Proof.* The first property follows by noting that $A_y = \alpha_y \mathcal{N}(y\rho, 1 - \rho^2) + (1-\alpha_y)Q_y = U_\rho(\alpha_y \delta_y + (1-\alpha_y)F_y)$ and using Fact A.1. This implies that for all $i \leq 2m$ we have that

$$\mathop{\mathbf{E}}_{X\sim U_\rho(\alpha_y\delta_y+(1-\alpha_y)F_y)}[h_i(X)] = \rho^i \mathop{\mathbf{E}}_{X\sim\alpha_y\delta_y+(1-\alpha_y)F_y}[h_i(X)] = \rho^i \mathop{\mathbf{E}}_{X\sim\mathcal{N}(0,1)}[h_i(X)] = \mathop{\mathbf{E}}_{X\sim\mathcal{N}(0,1)}[h_i(X)],$$

where the last equation uses that $\mathbf{E}_{X\sim\mathcal{N}(0,1)}[h_i(X)] = 0$ for $i > 0$ and $\mathbf{E}_{X\sim\mathcal{N}(0,1)}[h_0(X)] = 1$. Since $\{h_i : i \in [2m]\}$ form a basis of $\mathcal{P}(2m)$, the space of all polynomials of degree at most $2m$, it follows that $A_y$ continues to matches $2m$ moments with $\mathcal{N}(0, 1)$.

The $\chi^2$ bound is due to the bounded support in $[-B, B]$ and the Gaussian smoothing operation and can be shown as follows. First, we need the following fact whose proof is included in Appendix G for completeness.

**Fact B.2.** *For any one-dimensional distribution $P$ that matches the first $m$ moments with $\mathcal{N}(0, 1)$ and has $\chi^2(P, \mathcal{N}(0, 1)) < \infty$ the following identity is true:*

$$\chi^2(P, \mathcal{N}(0, 1)) = \sum_{i=m+1}^{\infty} \left( \mathop{\mathbf{E}}_{X\sim P}[h_i(X)] \right)^2 \ .$$

Let $M_y$ denote the distribution $\alpha_y \delta_y + (1 - \alpha_y)F_y$, i.e., the mixture before applying the Ornstein-Uhlenbeck operator. In order to apply Fact B.2 to $M_y$, we need to argue that its $\chi^2$-divergence is finite. Note that $F_y$ is a discrete distribution, the $U_\rho$ operator will transform it to a finite sum of Gaussians with variances strictly less than 2. We defer the proof of the following claim to Appendix G.

**Claim B.3.** *If $P = \sum_{i=1}^{k} \lambda_i N(\mu_i, \sigma_i^2)$ with $\mu_i \in \mathbb{R}$, $\sigma_i < \sqrt{2}$ and $\lambda_i \geq 0$ such that $\sum_{i=1}^{k} \lambda_i = 1$, we have that $\chi^2(P, \mathcal{N}(0, 1)) < \infty$.*

Using the formula of Fact B.2 and Fact A.1 for the individual terms, we get that

$$\chi^2(A_y, \mathcal{N}(0,1)) = \sum_{i=2m+1}^{\infty} \mathop{\mathbf{E}}_{X \sim U_\rho M_y}[h_i(X)]^2 = \sum_{i=2m+1}^{\infty} \rho^{2i} \mathop{\mathbf{E}}_{X \sim M_y}[h_i(X)]^2$$

$$= \sum_{i=2m+1}^{\infty} \rho^{2i} \left( \alpha_y h_i(y) + (1 - \alpha_y) \mathop{\mathbf{E}}_{x \sim F_y}[h_i(X)] \right)^2$$

$$\leq 2\alpha_y^2 \sum_{i=2m+1}^{\infty} \rho^{2i} h_i^2(y) + 2(1 - \alpha_y)^2 \sum_{i=2m+1}^{\infty} \rho^{2i} \mathop{\mathbf{E}}_{x \sim F_y}[h_i(X)]^2 , \qquad (2)$$

where the inequality uses that $(a + b)^2 \leq 2(a^2 + b^2)$ for all $a, b \in \mathbb{R}$. To bound this expression from above we will use the following tail bound for Hermite polynomials.

**Lemma 3.6.** *Let $h_i$ be the $i$-th normalized probabilist's Hermite polynomial. Then* $\max_{x \in \mathbb{R}} h_k^2(x) e^{-x^2/2} = O(k^{-1/6})$.

More details on how Lemma 3.6 follows from the result of [Kra04] can be found in Section C.3. For the first term of Equation (2), we have that

$$\sum_{i=2m+1}^{\infty} \rho^{2i} \alpha_y^2 h_i^2(y) \leq \sum_{i=2m+1}^{\infty} \rho^{2i} \alpha e^{-y^2 + \alpha y^2} O(e^{y^2/2})$$

$$\leq \alpha O(e^{y^2(\alpha - 1/2)}) \sum_{i=2m+1}^{\infty} \rho^{2i}$$

$$\leq \alpha O(e^{y^2(\alpha - 1/2)}) \rho^{2(2m+1)} / (1 - \rho^2) ,$$

where the first inequality uses Lemma 3.6 and the definition of $\alpha_y$. For the second term, we use the bounded support of $F_y$ in $[-B, B]$ along with the bound of Lemma 3.6 to obtain

$$\sum_{i=2m+1}^{\infty} \rho^{2i} \mathop{\mathbf{E}}_{x \sim F_y}[h_i(X)]^2 \leq \sum_{i=2m+1}^{\infty} \rho^{2i} O(e^{B^2/2}) \leq O(e^{B^2/2}) \sum_{i=2m+1}^{\infty} \rho^{2i} \leq O(e^{B^2/2}) \frac{\rho^{2(2m+1)}}{1 - \rho^2} .$$

This completes the proof of Lemma 2.6. $\qquad \square$

We include a detailed proof of Theorem 2.1 here for completeness.

*Proof of Theorem 2.1.* We will show that the following search problem $\mathcal{Z}$ has large statistical dimension: $\mathcal{D}$ is the set of distributions of the form $E_v(x, y) = \alpha D_v(x, y) + (1 - \alpha) N_v(x, y)$ for every $v \in \mathcal{S}^{d-1}$ and noise distribution $N_v$ as in Lemma 2.4. The reference distribution $R$ is $R = \mathcal{N}(0, I_d) \times \mathcal{N}(0, 1/\alpha)$. Let $\beta(v) = \rho v$ denote the regression vector corresponding to $E_v$. The set of solutions $\mathcal{F}$ is the set of all lists of size $\ell$ containing vectors of norm $\rho$ in $\mathbb{R}^d$ and the solution set $\mathcal{Z}(E_v)$ for the distribution $E_v$ is exactly the set of lists from $\mathcal{F}$ having at least one element $u$ at distance $\|u - \beta(v)\|_2 \leq \rho/2$. The appropriate subset of $\mathcal{D}$ that we will consider is the one corresponding to the set $S$ of nearly orthogonal vectors of Lemma 1.14, $\mathcal{D}_R = \{E_v\}_{v \in S}$.

Note that for any vector $u \in \mathbb{R}^d$ with norm $\rho$, there exists at most one element $E_v$ in $\mathcal{D}_R$ that satisfies $\|u - \beta(v)\|_2 \leq \rho/2$, since if there exists another $v'$ with $\|u - \beta(v')\|_2 \leq \rho/2$, then by triangle inequality $\|\beta(v) - \beta(v')\|_2 \leq \rho$. However, this cannot happen because $|v^T(v')| \leq O(d^{c-1/2})$ for all $v, v' \in S$ together with $d = 2^{\Omega(1/(1/2-c))}$ implies that $\|\beta(v) - \beta(v')\|_2 \geq \rho\sqrt{2(1 - v^T(v'))} \geq \rho$. This implies that for any solution list $L$, $|\mathcal{D}_R \setminus \mathcal{Z}^{-1}(L)| \geq |\mathcal{D}_R| - \ell$. We choose $\ell = |\mathcal{D}_R|/2$. We now calculate the pairwise correlation of the set $\mathcal{D}_R$. Let a pair of $u, v \in \mathcal{S}^{d-1}$.

$$\chi_{R(x,y)}(E_v(x,y), E_u(x,y)) = \mathop{\mathbf{E}}_{y \sim R(y)} \left[ \chi_{\mathcal{N}(0,I_d)}(E_v(x|y), E_u(x|y)) \right]$$

$$\leq |u^T v|^{2m+1} \mathop{\mathbf{E}}_{y \sim R(y)} \left[ \chi^2(A_y, \mathcal{N}(0,1)) \right]$$

$$= |u^T v|^{2m+1} \left( O(e^{B^2/2})/(1 - \rho^2) + \int_{\mathbb{R}} \alpha O(e^{y^2(\alpha - 1/2)}) \sqrt{\alpha} e^{-y^2 \alpha} dy \right)$$

$$\leq |u^T v|^{2m+1} O(e^{B^2/2})/(1-\rho^2)$$
$$\leq \Omega(d)^{-(2m+1)(1/2-c)} O(e^{B^2/2})/(1-\rho^2),$$

where the first line is due to Lemma 2.2, the second line is from Lemma 1.13 along with the observation that $E_v(x|y)$ is of the form $P_{A_y,v}$, the third line comes from the second part of Lemma 2.6, and the last one uses Lemma 1.14. Thus, by recalling that we can choose $B = C_2\sqrt{m}$ for a sufficiently large constant $C_2$, the set $\mathcal{D}_R$ is $(\gamma, b)$-correlated with respect to $R$, where $\gamma := \Omega(d)^{-(2m+1)(1/2-c)} e^{O(m)}/(1-\rho^2)$ and $b := e^{O(m)}/(1-\rho^2)$. The proof is concluded by applying Lemma 1.11 with $\gamma' = \gamma$. $\qquad\square$

We conclude this section with a note on the model and existing algorithmic results (extending the relevant discussion of Section 1.1).

**Remark B.4** (Comparison of SQ Lower Bound to Existing Upper Bounds). We remark that the model used in Theorem 1.5 (i.e., having a regressor with norm at most one and additive noise with small variance) is considered in both recent works [KKK19, RY20a] that provided list-decoding algorithms for the problem. In particular, these works give the following upper bounds:

- [KKK19] considers the model where $\|\beta\|_2 \leq 1$ and gives an algorithm that for every $\epsilon > 0$, runs in time $(d/(\alpha\epsilon))^{O\left(\frac{1}{\alpha^8\epsilon^8}\right)}$ and outputs a list of size $O(1/\alpha)$ containing a $\widehat{\beta}$ such that $\|\widehat{\beta} - \beta\|_2 \leq O(\sigma/\alpha) + \epsilon$. Note that this guarantee is better than the trivial upper bound of $1$ only if $\sigma = O(\alpha)$. To achieve error $1/4$, this algorithm runs in time $(d/\alpha)^{O\left(\frac{1}{\alpha^8}\right)}$. On the other hand, our lower bound for the complexity of any SQ algorithm becomes $\alpha d^{\Omega\left(1/\sqrt{\alpha}\right)}$.

- [RY20a] does not impose any constraint on $\|\beta\|_2$ and gives an algorithm that runs in time $(\|\beta\|_2/\sigma)^{\log(1/\alpha)} d^{O(1/\alpha^4)}$ and outputs a list of size $O((\|\beta\|_2/\sigma)^{\log(1/\alpha)})$ including a $\widehat{\beta}$ with the guarantee that $\|\widehat{\beta} - \beta\|_2 \leq O(\sigma/\alpha^{3/2})$. For the special case where $\|\beta\|_2 \leq 1$ (and $\sigma = O(\alpha^{3/2})$ in order for the error guarantee to be meaningful), this algorithm can achieve error $1/4$ in time $(1/\alpha^{3/2})^{\log(1/\alpha)} d^{O(1/\alpha^4)}$. In comparison, our lower bound becomes $\alpha^{3/2} d^{\Omega\left(1/\sqrt{\alpha}\right)}$.

# C Omitted Proofs from Section 3

## C.1 Proof of Proposition 3.2

We restate and prove the following proposition:

**Proposition 3.2.** *Fix $y \in \mathbb{R}$, $\alpha_y \in (0,1)$, $B > 0$, and $m \in \mathbb{Z}_+$. There exists a discrete distribution $F_y$ supported on at most $2m+1$ points in $[-B, B]$ such that $\alpha_y \delta_y + (1-\alpha_y)F_y$ matches the first $2m$ moments with $\mathcal{N}(0,1)$ if and only if $\mathbf{E}_{X \sim \mathcal{N}(0,1)}[p(X)] \geq \alpha_y p(y)$ for all $p \in \mathcal{P}^{\geq 0}(2m, B)$.*

We require the following result stating that for every distribution $Q$ with bounded support, there exists a discrete distribution $P$ with bounded support that matches the low-degree moments of $Q$.

**Lemma C.1.** *Let $B > 0$, $k \in \mathbb{Z}_+$, and $Q$ be any distribution with support in $[-B, B]$. Then there exists a discrete distribution $P$ with the following properties: (i) the support of $P$ is contained in $[-B, B]$, (ii) the first $k$ moments of $P$ agree with the first $k$ moments of $Q$, and (iii) $P$ is supported on at most $k+1$ points.*

*Proof.* Let $\mathcal{Q}$ be the set of distributions on $\mathbb{R}$ that are supported in $[-B, B]$ and let $\mathcal{Q}' \subset \mathcal{Q}$ be the set of Dirac delta distributions supported in $[-B, B]$, i.e., $\mathcal{Q}' = \{\delta_y : y \in [-B, B]\}$. Let $\mathcal{C} \subset \mathbb{R}^k$ and $\mathcal{C}' \subset \mathbb{R}^k$ be the set of all vectors $(x_1, \ldots, x_k)$ whose coordinates $x_1, \ldots, x_k$ are the moments of a distribution in $\mathcal{Q}$ and $\mathcal{Q}'$ respectively, i.e.,

$$\mathcal{C} := \{x \in \mathbb{R}^k : \exists Q \in \mathcal{Q} : \forall i \in [k], x_i = \mathop{\mathbf{E}}_{X \sim Q}[X^i]\},$$

$$\mathcal{C}' := \{x \in \mathbb{R}^k : \exists Q' \in \mathcal{Q}' : \forall i \in [k], x_i = \mathop{\mathbf{E}}_{X \sim Q'}[X^i]\}.$$

Note that there is a bijection between $\mathcal{C}'$ and $\mathcal{Q}'$. We now recall the following classical result stating convexity properties of $\mathcal{C}$ and its relation with $\mathcal{C}'$. We say a set $M$ is a convex hull of a set $M'$ if every $x \in M$ can be written as $x = \sum_{i=1}^{j} \lambda_i y_i$, where $j \in \mathbb{Z}_+$, $\sum_{i=1}^{j} \lambda_i = 1$, and for all $i \in [j]$: $\lambda_i \geq 0$, $y_i \in M'$.

**Lemma C.2** (Theorem 7.2 and 7.3 of [KS53]). *$\mathcal{C}$ is convex, closed, and bounded. Moreover, $\mathcal{C}$ is a convex hull of $\mathcal{C}'$.*

Let $x^* = (x_1^*, \ldots, x_k^*)$ be the first $k$ moments of $Q$. Since $x^* \in \mathcal{C}$, Caratheodory theorem and Lemma C.2 implies that $x^*$ can be written as a convex combination of at most $k + 1$ elements of $\mathcal{C}'$. This implies that there is a distribution, which is a convex combination of at most $k + 1$ Dirac delta distributions in $\mathcal{Q}'$, that matches the first $k$ moments with $x^*$. This completes the proof. $\qquad\square$

We can now prove the main result of this section.

*Proof of Proposition 3.2.* Let $X \sim \mathcal{N}(0, 1)$. We note that $F_y$ should have the moment sequence $x = (x_1, \ldots, x_{2m})$ where $x_i = (\mathbf{E}_{X \sim \mathcal{N}(0,1)}[X^i] - \alpha_y y^i)/(1 - \alpha_y)$ for $i \in [2m]$. Theorem 3.1 implies that this happens if and only if for all $p = (p_0, \ldots, p_{2m}) \in \mathcal{P}^{\geq 0}(2m, B)$, we have that $\sum_{i=0}^{2m} x_i p_i \geq 0$. The desired expression follows by noting that $\sum_{i=0}^{2m} x_i p_i = (\sum_{i=0}^{2m} p_i \mathbf{E}_{X \sim \mathcal{N}(0,1)}[X^i] - \alpha_y p_i y^i)/(1 - \alpha_y) = (\mathbf{E}_{X \sim \mathcal{N}(0,1)}[p(X)] - \alpha_y p(y))/(1 - \alpha_y)$. The result that $F_y$ is discrete follows from Lemma C.1. $\qquad\square$

## C.2 Proof of Lemma 3.3

For convenience, we restate the lemma below.

**Lemma 3.3.** *Let $m \in \mathbb{Z}_+$. If $B \geq C\sqrt{m}$ for some sufficiently large constant $C > 0$, then for every $q \in \mathcal{P}(m)$, it holds that $B^2 \mathbf{E}_{X \sim \mathcal{N}(0,1)}[q^2(X)] \geq 2\mathbf{E}_{X \sim \mathcal{N}(0,1)}[X^2 q^2(X)]$.*

*Proof.* Let $X \sim \mathcal{N}(0, 1)$. We can assume that $q$ is a non-zero polynomial. Then it suffices to bound $B$ from above by $\sqrt{2}$ times the following expression:

$$
\sup_{q \in \mathcal{P}(m), q \not\equiv 0} \sqrt{\frac{\mathbf{E}[X^2 q^2(X)]}{\mathbf{E}[q^2(X)]}} \leq \sup_{q \in \mathcal{P}(m), q \not\equiv 0} \sqrt{\frac{(\mathbf{E}[(X^2)^{m+1}])^{1/(m+1)} \left(\mathbf{E}[(q^2(X))^{\frac{m+1}{m}}]\right)^{\frac{m}{m+1}}}{\mathbf{E}[q^2(X)]}}
$$

$$
= \sup_{q \in \mathcal{P}(m), q \not\equiv 0} \frac{\|X\|_{L^{2m+2}} \|q(X)\|_{L^{\frac{2m+2}{m}}}}{\|q(X)\|_{L^2}},
$$

where the first step uses Hölder's inequality. Using standard concentration bounds for the standard Gaussian (or Fact 3.5 with $p(x) = x$), we get that $\|X\|_{L^{2m+2}} = O(\sqrt{m})$. Gaussian Hypercontractivity (Fact 3.5) implies that for any polynomial of degree at most $m$ and $r > 2$, $\|q(X)\|_{L^r} \leq (r-1)^{m/2} \|q(X)\|_{L^2}$. For $r = (2m + 2)/m$, we get that

$$
\frac{\|q(X)\|_{L^{\frac{2m+2}{m}}}}{\|q(X)\|_{L^2}} \leq \left(\frac{2m+2}{m} - 1\right)^{\frac{m}{2}} = \left(1 + \frac{2}{m}\right)^{\frac{m}{2}} \leq \exp(1).
$$

Therefore, $B \geq C\sqrt{m}$ suffices for a sufficiently large constant $C$. $\qquad\square$

## C.3 Proof of Lemma 3.4

We restate and prove the following:

**Lemma 3.4.** *Let $y \in \mathbb{R}$, $\alpha \in (0, 1/2)$, $m \in \mathbb{Z}_+$, and $\alpha_y = \sqrt{\alpha} \exp(-y^2(1 - \alpha)/2)$. Suppose $m \leq C/\sqrt{\alpha}$ for some sufficiently small constant $C > 0$. Then for all $r \in \mathcal{P}(m), r \not\equiv 0$: $r^2(y)/(\mathbf{E}_{X \sim \mathcal{N}(0,1)}[r^2(X)]) \leq 1/(2\alpha_y)$.*

We first recall the result on the tails of Hermite polynomials.

**Lemma 3.6.** *Let $h_i$ be the $i$-th normalized probabilist's Hermite polynomial. Then $\max_{x \in \mathbb{R}} h_k^2(x) e^{-x^2/2} = O(k^{-1/6})$.*

For completeness, we give an explicit calculation that translates the result of [Kra04] in our setting.

*Proof of Lemma 3.6.* We will split the analysis in two cases. First suppose the case when $k < 6$. As $h_k(\cdot)$ is a constant degree polynomial, we get that $\max_{x \in \mathbb{R}} h_k^2(x) \exp(-x^2/2)$ is a constant. For the rest of the proof, we will assume that $k \geq 6$.

For brevity, we will only consider the case where $k$ is even. The case where $k$ is odd is similar. Let $H_k(\cdot)$ be the physicist's Hermite polynomial. Recall that we can relate $h_k(\cdot)$ with $H_k(\cdot)$ with the following change of variable: $H_k(x) = \sqrt{2^k k!} h_k(\sqrt{2}x)$.

[Kra04, Theorem 1] implies the following:

$$\max_{x \in \mathbb{R}} \left( (H_k(x))^2 e^{-x^2} \right) = O\left( \sqrt{k} k^{-1/6} \binom{k}{0.5k} k! \right) . \tag{3}$$

From Equation (3) we obtain:

$$\max_{x \in \mathbb{R}} 2^k k! h_k^2(\sqrt{2}x) e^{-x^2} = \max_{x \in \mathbb{R}} 2^k k! h_k^2(x) e^{-x^2/2} = O\left( \sqrt{k} k^{-1/6} \binom{k}{0.5k} k! \right) .$$

This implies the following:

$$\max_{x \in \mathbb{R}} h_k^2(x) e^{-x^2/2} = O\left( k^{-1/6} \sqrt{k} \binom{k}{0.5k} 2^{-k} \right) = O(k^{-1/6}),$$

where we use that $\binom{k}{0.5k} 2^{-k}/\sqrt{k} = O(1)$. $\square$

*Proof of Lemma 3.4.* Let $h_i$ be the $i$-th normalized probabilist's Hermite polynomial. Since $r$ is a polynomial of degree at most $m$ and $\{h_i, i \in [m]\}$ form a basis for $\mathcal{P}(m)$, we can represent $r(x) = \sum_{i=1}^m a_i h_i(x)$ for some $a_i \in \mathbb{R}$. Using orthonormality of $h_i$ under the Gaussian measure, we get that $\mathbf{E}_{X \sim \mathcal{N}(0,1)}[r^2(X)] = \sum_{i=1}^m a_i^2$. Since $r$ is a non-zero polynomial, we have that $\sum_{i=1}^m a_i^2 > 0$. We thus have that

$$\sup_{r \in \mathcal{P}(m), r \not\equiv 0} \frac{r^2(y)}{\mathbf{E}_{X \sim \mathcal{N}(0,1)}[r^2(X)]} = \sup_{a_1,\ldots,a_m \in \mathbb{R}, \sum_{i=1}^m a_i^2 > 0} \frac{\sum_{i=1}^m \sum_{j=1}^m a_i a_j h_i(y) h_j(y)}{\sum_{i=1}^m a_i^2}$$

$$= \sup_{a_1,\ldots,a_m \in \mathbb{R}, \sum_{i=1}^m a_i^2 > 0} \frac{\sqrt{\sum_{i=1}^m \sum_{j=1}^m a_i^2 a_j^2} \sqrt{\sum_{i=1}^m \sum_{j=1}^m h_i^2(y) h_j^2(y)}}{\sum_{i=1}^m a_i^2}$$

$$= \sum_{i=1}^m h_i^2(y).$$

Therefore, we need to show that, for all $y \in \mathbb{R}$, $\sum_{i=1}^m \alpha_y h_i^2(y) \leq 1/2$ whenever $m \leq C/\sqrt{\alpha}$ for a sufficiently small constant $C > 0$. We will now split the analysis in two cases:

**Case 1:** $|y| \leq 1/\sqrt{\alpha}$. Using Lemma 3.6 and the assumption that $|y|^2 \alpha \leq 1$, we can bound the desired expression as follows:

$$\max_{|y| \leq 1/\sqrt{\alpha}} \alpha_y h_i^2(y) = \max_{|y| \leq 1/\sqrt{\alpha}} \sqrt{\alpha} \exp(y^2 \alpha/2) \exp(-y^2/2) h_i^2(y)$$

$$\leq \sqrt{\alpha e} \sup_{y \in \mathbb{R}} \exp(-y^2/2) h_i^2(y)$$

$$= O(\sqrt{\alpha} i^{-1/6}).$$

Therefore, we get the following bound on $\sum_i h_i^2(y)$.

$$\sum_{i=1}^m \alpha_y h_i^2(y) = O\left( \sqrt{\alpha} \sum_{i=1}^m i^{-1/6} \right) = O(\sqrt{\alpha} m^{5/6}) .$$

The last expression is less than $1/2$ when $m = O(1/\alpha^{3/5})$.

**Case 2:** $|y| \geq 1/\sqrt{\alpha}$. We will use rather crude bounds here. We have the following explicit expression of $h_i(\cdot)$ (see, for example, [AAR99, Sze89]):

$$|h_i(x)| = \left| \frac{He_i(x)}{\sqrt{i!}} \right| = \left| \sqrt{i!} \sum_{j=0}^{\lfloor i/2 \rfloor} \frac{(-1)^j}{j!(i-2j)!} \frac{x^{i-2j}}{2^j} \right| = \left| \sqrt{i!} x^i \sum_{j=0}^{\lfloor i/2 \rfloor} \frac{(-1)^j}{(2j)!(i-2j)!} x^{-2j} \frac{(2j)!}{j! 2^j} \right|$$

$$\leq \sqrt{i!} |x|^i \sum_{k=0}^{i} \frac{i!}{k!(i-k)!} |x|^{-k} \leq (i|x|)^i (1 + |x|^{-1})^i = i^i (1 + |x|)^i.$$

Therefore, we get the following relation for all $|y| > 1$, $\alpha < 0.5$, and $i \in \mathbb{N}$:

$$\alpha_y h_i^2(y) = \sqrt{\alpha} \exp(-y^2(1-\alpha)/2) h_i^2(y)$$
$$\leq \sqrt{\alpha} \exp(-y^2/4)(2i|y|)^{2i}$$
$$= \sqrt{\alpha} \exp(-y^2/4 + 2i \log(2i|y|)).$$

The expression above is at most $C'\sqrt{\alpha}$ for a constant $C' > 0$ for all $|y| \geq c'\sqrt{i \log i}$ for a constant $c' > 0$. The latter condition holds whenever $1/\sqrt{\alpha} \geq c'\sqrt{i \log i}$. It suffices that $i = O(1/\alpha^{0.9})$. Overall, we get the following bound when $m = O(1/\alpha^{0.9})$:

$$\sup_{|y| > 1/\sqrt{\alpha}} \sum_{i=1}^{m} \alpha_y h_i^2(y) = O(\sqrt{\alpha}m).$$

The last expression is less than $1/2$ when $m \leq C/\sqrt{\alpha}$ for some constant $C > 0$. This completes the proof of Lemma 3.4. □

## D  Information-Theoretic Bounds

### D.1  Upper Bound on Sample Complexity

In this section, we show that $n = \text{poly}(d, 1/\alpha)$ samples suffice for list-decodable linear regression. We note that the similar guarantees can be achieved by modifying the analysis in [KKK19] although the sample complexity details are not explicit there. At a high level, both Theorem D.1 and [KKK19] rely on the following properties of inliers: anti-concentration of covariates $(X)$ and concentration of additive noise $(\eta)$.

**Theorem D.1.** *There is a (computationally inefficient) algorithm that uses $O(d/\alpha^3)$ samples from a $(1-\alpha)$-corrupted version of a Gaussian linear regression model of Definition 1.2 with $S' = \mathbb{R}^d$, and returns a list $\mathcal{L}$ of $|\mathcal{L}| = O(1/\alpha)$ many hypotheses such that with high probability at least one of them is within $\ell_2$-distance $O((\sigma/\alpha)\sqrt{\log(1/\alpha)})$ from the regression vector.*

The proof strategy is similar to [DKS18]. When $S$ is a set, we use the notation $X \sim_u S$ to denote that $X$ is distributed according to the uniform distribution on $S$. We require the following theorem:

**Theorem D.2** (VC Inequality). *Let $\mathcal{F}$ be a class of Boolean functions with finite VC dimension $\text{VC}(\mathcal{F})$ and let a probability distribution $D$ over the domain of these functions. For a set $S$ of $n$ independent samples from $D$*

$$\sup_{f \in \mathcal{F}} \left| \Pr_{X \sim_u S}[f(X)] - \Pr_{X \sim D}[f(X)] \right| \lesssim \sqrt{\frac{\text{VC}(\mathcal{F})}{n}} + \sqrt{\frac{\log(1/\tau)}{n}},$$

*with probability at least $1 - \tau$.*

*Proof of Theorem D.1.* Recall the notation in Definitions 1.1 and 1.2. Let $T$ be the set of points generated from the $(1-\alpha)$-corrupted version of $D_{\beta^*}$ for some unknown $\beta^* \in \mathbb{R}^d$. Let $S_1$ be the set of points that are sampled from $D_{\beta^*}$. Since inliers are sampled with probability $\alpha$, we have that $|S_1| \geq \alpha|T|/2$ with high probability. For a $t \geq 0$, define $\mathcal{H}_{t,\gamma}$ as follows:

$$\mathcal{H}_{t,\gamma} := \left\{ \beta \in \mathbb{R}^d : \exists T' \subset T, \ |T'| = \alpha|T|/2, \right. \tag{4}$$

$$\Pr_{(X,y)\sim_u T'}[|y - X^T\beta| > \sigma t] \le \alpha/20, \tag{5}$$

$$\left.\forall v \in \mathcal{S}^{d-1}, \gamma' \ge \gamma: \quad \Pr_{(X,y)\sim_u T'}[|y - X^T\beta - \gamma'v^T X| \le \sigma t] \le \alpha/20\right\}. \tag{6}$$

Recall that the distribution of inliers is $X \sim \mathcal{N}(0, I_d)$ and $y = X^T\beta^* + \eta$, where $\eta \sim \mathcal{N}(0, \sigma^2)$ independent of $X$. If $|S_1| \ge Cd/\alpha^2$ for a sufficiently large constant $C$, then we claim that $\beta^* \in \mathcal{H}_{t,\gamma}$ with $t = \Theta(\sqrt{\log(1/\alpha)})$ and $\gamma = 40\sigma t/\alpha = \Theta((\sigma/\alpha)\sqrt{\log(1/\alpha)})$. Let $S'$ be a set of i.i.d. points sampled from $D_{\beta^*}$ with $|S'| = |T|\alpha/2$. We first argue that conditions (5) and (6) hold under $(X,y) \sim D_{\beta^*}$, even after replacing $\alpha/20$ with $\alpha/40$ in conditions (5) and (6), with the claimed bounds on $t$ and $\gamma$, and then the required result on $(X,y) \sim_u S'$ will follow from the VC inequality. Since $y - X^T\beta^* \sim \mathcal{N}(0, \sigma^2)$ under $D_{\beta^*}$, we get that $\mathbf{Pr}[|y - X^T\beta^*| > \sigma t] \le \alpha/40$ because of Gaussian concentration. Let $G \sim \mathcal{N}(0,1)$ independent of $\eta$. For condition (6), the expression again reduces to concentration of a Gaussian distribution:

$$\Pr_{\eta\sim\mathcal{N}(0,\sigma^2), G\sim\mathcal{N}(0,1)}[|\eta + \gamma'G| \le \sigma t] = \Pr_{Z\sim\mathcal{N}(0,\sigma^2+\gamma'^2)}[|Z| \le \sigma t] \lesssim \frac{\sigma t}{\gamma'},$$

which is less than $\alpha/40$ for $\gamma' \ge \gamma = 40t\sigma/\alpha$. The desired conclusion now follows by noting that conditions (5) and (6) follow by uniform concentration of linear threshold functions on $(X,y)$, which have VC dimension $O(d)$ and the condition that $|S'| = \Omega(d/\alpha^2)$.

We then show that any $(2\gamma)$-packing of the set $\mathcal{H}_{t,\gamma}$ has size $O(1/\alpha)$. Having this, it follows that there exists a $(2\gamma)$-cover of size $O(1/\alpha)$ and the output of the algorithm, $\mathcal{L}$, consists of returning any such cover. The key claim for bounding the size of any $(2\gamma)$-packing is that the pairwise intersection between the sets $T'$ from condition (4) are small.

**Claim D.3.** *Let $\beta_1, \ldots, \beta_k \in \mathcal{H}_{t,\gamma}$ such that $\|\beta_i - \beta_j\|_2 > 2\gamma$ for all $i, j \in [k]$ and $i \ne j$. Let $T'_i$ be the corresponding subsets of $T$ satisfying the condition* (4). *Then $|T'_i \cap T'_j| \le \alpha(|T'_i| + |T'_j|)/20$.*

*Proof.* Fix an $i \ne j$. Let $\beta_i - \beta_j = 2v\gamma'$, where $v \in \mathcal{S}^{d-1}$ and $\gamma' \ge \gamma$. Let $\mathcal{E}$ be the event $\{(X,y): |y - X^T\beta_j| \le \sigma t\}$ and $\mathcal{E}^c$ be its complement. As $T'_i$ and $T'_j$ are sets of size $\alpha|T|/2$, we have that

$$|T'_i \cap T'_j| = \frac{|T'_i| + |T'_j|}{2}\left(\frac{|T'_i \cap T'_j \cap \mathcal{E}|}{|T'_i|} + \frac{|T'_i \cap T'_j \cap \mathcal{E}^c|}{|T'_j|}\right)$$

$$\le \frac{|T'_i| + |T'_j|}{2}\left(\frac{|T'_i \cap \mathcal{E}|}{|T'_i|} + \frac{|T'_j \cap \mathcal{E}^c|}{|T'_j|}\right) = \frac{|T'_i| + |T'_j|}{2}\left(\Pr_{(X,y)\sim_u T'_i}[\mathcal{E}] + \Pr_{(X,y)\sim_u T'_j}[\mathcal{E}^c]\right).$$

As $\beta_j \in \mathcal{H}_{t,\gamma}$, we have that $\mathbf{P}_{(X,y)\sim_u T'_j}[\mathcal{E}^c] \le \alpha/20$ by condition (5). We now bound the first term.

$$\Pr_{(X,y)\sim_u T'_i}[\mathcal{E}] = \Pr_{(X,y)\sim_u T'_i}[|y - X^T\beta_i - \gamma v^T X| \le \sigma t],$$

which is less than $\alpha/20$ by the condition (6). This completes the proof of the claim. $\square$

We use this to show that there cannot exist a $(2\gamma)$-packing of size $k \ge 4/\alpha$. To see this, assume that $k = 4/\alpha$, then

$$|T| \ge \sum_{i=1}^k |T'_i| - \sum_{1\le i<j\le k} |T'_i \cap T'_j| \ge \left(1 - \frac{\alpha}{20}(k-1)\right)\sum_{i=1}^k |T'_i| \ge \frac{4}{5}k\alpha\frac{|T|}{2} > |T|.$$

This yields a contradiction, completing the proof of Theorem D.1. $\square$

## D.2 Information-Theoretic Lower Bound on Error

We establish the following lower bound on the error of any list-decoding algorithm for linear regression.

**Theorem D.4.** *Let $0 < \alpha < 1/2$, $\sigma > 0$, $k > 1$ such that $k = O(1/(\alpha^2 \log(1/\alpha)))$, and $d \in \mathbb{Z}_+$ such that $d > (\log(1/\alpha^k))^C$, where $C$ is a sufficiently large constant. Any list-decodable algorithm that receives a $(1-\alpha)$-corrupted version of $D_\beta$ (defined in Definition 1.2) for some unknown $\beta \in \mathbb{R}^d$, and returns a list $\mathcal{L}$ of size $|\mathcal{L}| = O((1/\alpha)^k)$ has error bound $\Omega\left(\frac{\sigma}{\alpha\sqrt{k\log(1/\alpha)}}\right)$ with high probability.*

*Proof.* Let $\rho > 0$ to be decided later. We will take $\beta$ to be of the form $\rho v$ for some unit vector $v$. By abusing notation, let $D_v(x, y)$ be the joint distribution on $(X, y)$ from the linear model $X \sim \mathcal{N}(0, I_d)$, $y = \beta^T X + \eta$, where $\eta \sim \mathcal{N}(0, \sigma^2)$ independently of $X$ and $\beta = \rho v$. As $d$ is large enough, let $S'$ be a subset of the set $S$ of nearly orthogonal unit vectors of $\mathbb{R}^d$ from Lemma 1.14 with $|S'| = \lfloor 0.5(1/\alpha)^k \rfloor$ for $k > 1$. Consider the set of distributions $\{D_v\}_{v \in S'}$ and note that for every distinct pair $u, v \in S$ we have that $\|\rho u - \rho v\|_2 \geq c\rho$ for some $c > 0$. We want to show that after adding $(1 - \alpha)$-fraction of outliers these distributions become indistinguishable, i.e., there exists some distribution that is pointwise greater than $\alpha D_v$ for every $v \in S'$. This will lead to a lower bound on error of the form $\Omega(\rho)$. Let $P$ be the joint pseudo-distribution on $(X, y)$ such that $P(x, y) = \max_{v \in S} D_v(x, y)$ and denote by $\|P\|_1$ the normalizing factor $\int_\mathbb{R} \int_{\mathbb{R}^d} P(x, y)\mathrm{dxdy}$. We will show that $P/\|P\|_1 \geq \alpha D_v$ pointwise. To this end, it suffices to show that $\|P\|_1 \leq 1/\alpha$. Denote $z := v^T x$. Noting that $D_v$'s marginal on $x$ is $\mathcal{N}(0, I_d)$ and the conditional $D_v(y|x)$ is $\mathcal{N}(\rho z, \sigma^2)$ we can write

$$D_v(x, y) = \frac{1}{\sqrt{2\pi}\sigma} \exp\left(-\frac{|y - \rho z|^2}{2\sigma^2}\right) \frac{1}{(\sqrt{2\pi})^d} \exp\left(-\frac{\|x\|^2}{2}\right)$$

$$= \frac{1}{(\sqrt{2\pi})^{d+1}\sigma} \exp\left(-\frac{|y - \rho z|^2}{2\sigma^2} - \frac{\|x\|^2}{2}\right).$$

For some $\sigma_1$ to be defined later, take $R$ to be the reference distribution where $X \sim \mathcal{N}(0, I_d)$ and $y \sim \mathcal{N}(0, \sigma_1^2)$ independently. We now calculate the ratio of density of $R$ with $D_v$ at arbitrary $(x, y)$:

$$\frac{R(x, y)}{D_v(x, y)} = \frac{R(y)R(x|y)}{D_v(y)D_v(x|y)}$$

$$= \frac{\frac{1}{(\sqrt{2\pi})^{d+1}\sigma_1} \exp\left(-0.5\|x\|^2 - 0.5y^2/\sigma_1^2\right)}{\frac{1}{(\sqrt{2\pi})^{d+1}\sigma} \exp\left(-0.5\|x\|^2 - 0.5\frac{\rho^2}{\sigma^2}\left(z - \frac{y}{\rho}\right)^2\right)}$$

$$= \frac{\sigma}{\sigma_1} \exp\left(-\frac{y^2}{2\sigma_1^2} + \frac{\rho^2}{2\sigma^2}\left(z - \frac{y}{\rho}\right)^2\right)$$

$$\geq \frac{\sigma}{\sigma_1} \exp\left(-\frac{y^2}{2\sigma_1^2}\right).$$

We want that this expression is greater than $2\alpha$ with high probability under $D_v$. Under $D_v$, $y \sim \mathcal{N}(0, \sigma_y^2)$, with probability $1 - \alpha^{k-1}$, $|y| \leq 10\sqrt{k}\sigma_y\sqrt{\log(1/\alpha)}$. Setting $\sigma_1 = 10\sqrt{k}\sigma_y\sqrt{\log(1/\alpha)}$, we get that for $|y| \leq 10\sqrt{k}\sigma_y\sqrt{\log(1/\alpha)}$,

$$\frac{R(x, y)}{D_v(x, y)} \geq \frac{1}{100\sqrt{k}}\frac{\sigma}{\sigma_y\sqrt{\log(1/\alpha)}}. \tag{7}$$

We can now try to maximize $\rho$ (and thus $\sigma_y$) so that the expression on the right-hand side in (7) is greater than $2\alpha$. This holds as long as $\rho$ satisfies the following:

$$\sigma_y^2 = \sigma^2 + \rho^2 \leq \frac{\sigma^2}{C'k\alpha^2\log(1/\alpha)},$$

As $k = O(1/(\alpha^2\log(1/\alpha)))$, the condition above shows that $\rho$ can be as large as $\Theta((\sigma/(\sqrt{k}\alpha))/\sqrt{\log(1/\alpha)})$. Finally we show that $\|P\|_1$ is less than $1/\alpha$ as follows:

$$\|P\|_1 = \int_\mathbb{R} \int_{\mathbb{R}^d} P(x, y)\mathrm{dxdy}$$

$$= \int_{\mathbb{R}} \int_{\mathbb{R}^d} P(x,y)\mathbf{1}(|y| \leq 10\sqrt{k}\sigma_y\sqrt{\log(1/\alpha)})\mathrm{dxdy} + \int_{\mathbb{R}} \int_{\mathbb{R}^d} P(x,y)\mathbf{1}(|y| > 10\sqrt{k}\sigma_y\sqrt{\log(1/\alpha)})\mathrm{dxdy}$$

$$\leq \frac{1}{2\alpha} \int_{\mathbb{R}} \int_{\mathbb{R}^d} R(x,y)\mathrm{dxdy} + \int_{\mathbb{R}} \int_{\mathbb{R}^d} P(x,y)\mathbf{1}(|y| > 10\sqrt{k}\sigma_y\sqrt{\log(1/\alpha)})\mathrm{dxdy}$$

$$\leq \frac{1}{2\alpha} + \sum_{v \in S'} \Pr_{(X,y)\sim D_v} \left[ |y| > 10\sqrt{k}\sigma_y\sqrt{\log(1/\alpha)} \right]$$

$$\leq \frac{1}{2\alpha} + |S'|\alpha^{k-1} \leq 1/\alpha,$$

where the first inequality uses that for $|y| \leq 10\sqrt{k}\sigma_y\sqrt{\log(1/\alpha)})$, we have that $P(x,y)/R(x,y) = \max_v D_v(x,y)/R(x,y) \leq 1/2\alpha$ from (7), and the last inequality follows by noting that $|S'| \leq 0.5(1/\alpha)^k$. $\qquad\square$

# E    Hypothesis Testing Version of Robust Linear Regression

**Organization**    We introduce Problem E.2, which is the hypothesis testing variant of the search problem that we discussed in the main text of this paper. We first show the SQ hardness of Problem E.2 in Theorem E.3. In Section E.2, we give an efficient reduction from Problem E.2 to list-decodable linear regression, showing that Problem E.2 is indeed easier than the list-decodable linear regression problem.

We begin by formally defining a hypothesis problem.

**Definition E.1** (Hypothesis testing).  Let a distribution $D_0$ and a set $\mathcal{S} = \{D_u\}_{u \in S}$ of distributions on $\mathbb{R}^d$. Let $\mu$ be a prior distribution on the indices $S$ of that family. We are given access (via i.i.d. samples or oracle) to an *underlying* distribution where one of the two is true:

- $H_0$: The underlying distribution is $D_0$.

- $H_1$: First $u$ is drawn from $\mu$ and then the underlying distribution is set to be $D_u$.

We say that a (randomized) algorithm solves the hypothesis testing problem if it succeeds with non-trivial probability (i.e., greater than 0.9).

We now introduce the following hypothesis testing variant of the $(1-\alpha)$-contaminated linear regression problem:

**Problem E.2.**  Let $\alpha \in (0, 1/2)$, $\rho \in (0, 1)$. Let $S$ be the set of $d$-dimensional nearly orthogonal vectors from Lemma 1.14. We are given access (via i.i.d. samples or oracle) to an *underlying* distribution where one of the two is true:

- $H_0$: The underlying distribution is $R = \mathcal{N}(0, I_d) \times \mathcal{N}(0, 1/\alpha)$.

- $H_1$: First, a vector $v$ is chosen uniformly at random from $S$. The underlying distribution is set to be $E_v$, i.e., the $(1-\alpha)$-additively corrupted linear model of Definition 1.2 with $\beta = \rho v$, $\sigma^2 = 1 - \rho^2$, and a fixed noise distribution $N_v$ as specified in Lemma 2.4.

Using the reduction outlined in Lemma E.9, it follows that $O(d/\alpha^3)$ samples suffice to solve Problem E.2 when $\sigma \leq O(\alpha/\sqrt{\log(1/\alpha)})$. On the other hand, the following result shows an SQ lower bound of $d^{\mathrm{poly}(1/\alpha)}$.

**Theorem E.3** (SQ Hardness of Problem E.2).  *Let $0 < c < 1/2$, $m \in \mathbb{Z}_+$ with $m \leq c_1/\sqrt{\alpha}$ for some sufficiently small constant $c_1 > 0$ and $d = m^{\Omega(1/c)}$. Every SQ algorithm that solves Problem E.2 either performs $2^{\Omega(d^{c/4})}$ queries or performs at least one query to* $\mathrm{STAT}\left(\Omega(d)^{-(2m+1)(1/4-c/2)}e^{O(m)}/\sqrt{1-\rho^2}\right)$.

We note that the lower bound on the (appropriate) statistical dimension implies SQ hardness of the (corresponding) hypothesis testing problem. As the Problem E.2 differ slightly from the kind of hypothesis testing problems considered in [FGR+17], we provide the proof of Theorem E.3 in Section E.1, where we introduce the relevant statistical dimension from [BBH+20] (Definition E.4 in this paper). In Section F, we also show the hardness of Problem E.2 against low-degree polynomial tests.

### E.1 Hardness of Hypothesis Testing in SQ Model

We need the following variant of the statistical dimension from [BBH$^+$20], which is closely related to the hypothesis testing problems considered in this section. Since this is a slightly different definition from the statistical dimension (SD) used so far, we will assign the distinct notation (SDA) for it.

**Notation** For $f : \mathbb{R} \to \mathbb{R}$, $g : \mathbb{R} \to \mathbb{R}$ and a distribution $D$, we define the inner product $\langle f, g \rangle_D = \mathbf{E}_{X \sim D}[f(X)g(X)]$ and the norm $\|f\|_D = \sqrt{\langle f, f \rangle_D}$.

**Definition E.4** (Statistical Dimension). For the hypothesis testing problem of Definition E.1, we define the *statistical dimension* $\mathrm{SDA}(\mathcal{S}, \mu, n)$ as follows:

$$\mathrm{SDA}(\mathcal{S}, \mu, n) = \max \left\{ q \in \mathbb{N} : \mathop{\mathbf{E}}_{u,v\sim\mu}[|\langle \bar{D}_u, \bar{D}_v \rangle_{D_0} - 1| \mid \mathcal{E}] \leq \frac{1}{n} \text{ for all events } \mathcal{E} \text{ s.t. } \mathop{\mathbf{Pr}}_{u,v\sim\mu}[\mathcal{E}] \geq \frac{1}{q^2} \right\} .$$

We will omit writing $\mu$ when it is clear from the context.

**Theorem E.5** (Theorem A.5 of [BBH$^+$20]). *Let $\mathcal{S} = \{D_u\}_{u\in S}$ vs. $D_0$ be a hypothesis testing problem with prior $\mu$ on $\mathcal{S}$. If $\mathrm{SDA}(\mathcal{S}, \mu, 3/t) > q$, then every SQ algorithm that solves the hypothesis testing problem either makes at least $q$ queries, or makes at least one query to $\mathrm{STAT}(\sqrt{t})$.*

In order to prove Theorem E.2, we will prove a lower bound on the SDA of Problem E.2. As we will show later, Problem E.2 is a special case of the following hypothesis testing problem:

**Problem E.6** (Non-Gaussian Component Hypothesis Testing). Let $R$ be the joint distribution $R$ over the pair $(X, y) \in \mathbb{R}^{d+1}$ where $X \sim \mathcal{N}(0, I_d)$ and $y \sim R(y)$ independently of $X$. Let $E_v$ be the joint distribution over pairs $(X, y) \in \mathbb{R}^{d+1}$ where the marginal on $y$ is again $R(y)$ but the conditional distribution $E_v(x|y)$ is of the form $P_{A_y, v}$ (with $P_{A_y, v}$ as in Definition 1.12). Define $\mathcal{S} = \{E_v\}_{v \in S}$ for $S$ being the set of $d$-dimensional nearly orthogonal vectors from Lemma 1.14 and let the hypothesis testing problem be distinguishing between $R$ vs. $\mathcal{S}$ with prior $\mu$ being the uniform distribution on $S$.

The following lemma translates the $(\gamma, \beta)$-correlation of $\mathcal{S}$ to a lower bound for the statistical dimension of the hypothesis testing problem. The proof is very similar to that of Corollary 8.28 of [BBH$^+$20] but it is given below for completeness.

**Lemma E.7.** *Let $0 < c < 1/2$ and $d, m \in \mathbb{Z}_+$ such that $d = m^{\Omega(1/c)}$. Consider the hypothesis testing problem of Problem E.6 where for every $y \in \mathbb{R}$ the distribution $A_y$ matches the first $m$ moments with $\mathcal{N}(0, 1)$ and $\mathbf{E}_{y\sim R(y)}[\chi^2(A_y, \mathcal{N}(0, 1))] < \infty$. Then, for any $q \geq 1$,*

$$\mathrm{SDA}\left( \mathcal{D}, \frac{\Omega(d)^{(m+1)(1/2-c)}}{\mathbf{E}_{y\sim R(y)}[\chi^2(A_y, \mathcal{N}(0, 1))]\left( \frac{q^2}{2^{\Omega(d^{c/2})}} + 1 \right)} \right) \geq q .$$

*Proof.* The first part is to calculate the correlation of the set $\mathcal{S}$ exactly as we did in the proof of Theorem 2.1. By Lemma 1.14, Lemma 1.13 and Lemma 2.2 we know that the set $\mathcal{S}$ is $(\gamma, \beta)$-correlated with $\gamma = \Omega(d)^{-(m+1)(1/2-c)} \mathbf{E}_{y\sim R(y)}[\chi^2(A_y, \mathcal{N}(0, 1))]$ and $\beta = \mathbf{E}_{y\sim R(y)}[\chi^2(A_y, \mathcal{N}(0, 1))]$.

We next calculate the SDA according to Definition E.4. We denote by $\bar{E}_v$ the ratios of the density of $E_v$ to the density of $R$. Note that the quantity $\langle \bar{E}_u, \bar{E}_v \rangle - 1$ used there is equal to $\langle \bar{E}_u - 1, \bar{E}_v - 1 \rangle$. Let $\mathcal{E}$ be an event that has $\mathbf{Pr}_{u,v\sim\mu}[\mathcal{E}] \geq 1/q^2$. For $d$ sufficiently large we have that

$$\mathop{\mathbf{E}}_{u,v\sim\mu}[|\langle \bar{E}_u, \bar{E}_v \rangle - 1 | \mathcal{E}] \leq \min\left( 1, \frac{1}{|\mathcal{S}|\,\mathbf{Pr}[\mathcal{E}]} \right) \mathop{\mathbf{E}}_{y\sim R(y)}[\chi^2(A_y, \mathcal{N}(0, 1))]$$

$$+ \max\left( 0, 1 - \frac{1}{|\mathcal{S}|\,\mathbf{Pr}[\mathcal{E}]} \right) \frac{\mathbf{E}_{y\sim R(y)}[\chi^2(A_y, \mathcal{N}(0, 1))]}{\Omega(d)^{(m+1)(1/2-c)}}$$

$$\leq \mathop{\mathbf{E}}_{y\sim R(y)}[\chi^2(A_y, \mathcal{N}(0, 1))] \left( \frac{q^2}{2^{\Omega(d^c)}} + \frac{1}{\Omega(d)^{(m+1)(1/2-c)}} \right)$$

$$= \mathop{\mathbf{E}}_{y\sim R(y)}[\chi^2(A_y, \mathcal{N}(0, 1))] \frac{q^2 \Omega(d)^{(m+1)(1/2-c)} + 2^{\Omega(d^c)}}{2^{\Omega(d^c)}\Omega(d)^{(m+1)(1/2-c)}}$$

$$= \mathop{\mathbf{E}}_{y \sim R(y)}[\chi^2(A_y, \mathcal{N}(0,1))] \left( \frac{\Omega(d)^{(m+1)(1/2-c)}}{q^2 \Omega(d)^{(m+1)(1/2-c)}/2^{\Omega(d^c)} + 1} \right)^{-1}$$

$$= \mathop{\mathbf{E}}_{y \sim R(y)}[\chi^2(A_y, \mathcal{N}(0,1))] \left( \frac{\Omega(d)^{(m+1)(1/2-c)}}{q^2/2^{\Omega(d^{c/2})} + 1} \right)^{-1},$$

where the first inequality uses that $\mathbf{Pr}[u = v|\mathcal{E}] = \mathbf{Pr}[u = v, \mathcal{E}]/\mathbf{Pr}[\mathcal{E}]$ and bounds the numerator in two different ways: $\mathbf{Pr}[u = v, \mathcal{E}]/\mathbf{Pr}[\mathcal{E}] \leq \mathbf{Pr}[u = v]/\mathbf{Pr}[\mathcal{E}] = 1/(|\mathcal{D}|\mathbf{Pr}[\mathcal{E}])$ and $\mathbf{Pr}[u = v, \mathcal{E}]/\mathbf{Pr}[\mathcal{E}] \leq \mathbf{Pr}[\mathcal{E}]/\mathbf{Pr}[\mathcal{E}] = 1$. □

We note that the lemma above and Theorem E.5 show SQ hardness of Problem E.6. In the remainder of this section, we will apply these results to Problem E.2.

**Corollary E.8.** *Let $0 < c < 1/2$, $m \in \mathbb{Z}_+$ with $m \leq c_1/\sqrt{\alpha}$ for some sufficiently small constant $c_1 > 0$ and $d = m^{\Omega(1/c)}$. Consider the hypothesis testing problem of Problem E.2. Then, for any $k < d^{c/4}$:*

$$\mathrm{SDA}\left( \mathcal{D}, \frac{\Omega(d)^{(2m+1)(1/2-c)}}{e^{O(m)}/(1 - \rho^2)} \right) \geq 100^k .$$

*Proof.* We note that Problem E.2 is a special case of Problem E.6 (see Fact 2.3 and Lemma 2.4 which show that the conditional distributions are of the form $P_{A_y, v}$). In Lemma E.7 we use $q = \sqrt{2^{\Omega(d^{c/2})}(n/n')}$ with $n' = n = \frac{\Omega(d)^{(2m+1)(1/2-c)}}{\mathbf{E}_{y \sim R(y)}[\chi^2(A_y, \mathcal{N}(0,1))]}$ to get that $\mathrm{SDA}(\mathcal{D}, n) > 100^k$ for $k < d^{c/4}$. The first part of Lemma 2.6 states that the distributions $A_y$'s match the first $2m$ moments with $\mathcal{N}(0,1)$ for $m \leq c_1/\sqrt{\alpha}$ and the second part implies that $\mathbf{E}_{y \sim R(y)}[\chi^2(A_y, \mathcal{N}(0,1))] = O(e^m)/(1 - \rho^2)$. This completes the proof. □

We conclude by noting the hardness of Problem E.6 and thus Problem E.2 in the SQ model. The proof of Theorem E.3 follows from Corollary E.8 and Theorem E.5.

## E.2 Reduction from Hypothesis Testing to List-Decodable Regression

We now show that any list-decoding algorithm for robust linear regression can be efficiently used to solve Problem E.2, that is, hypothesis testing efficiently reduces to list-decodable estimation. For a list $\mathcal{L}$ and $i \in [|\mathcal{L}|]$, we use $\mathcal{L}(i)$ to denote the $i$-th element of $\mathcal{L}$.

**Lemma E.9.** *Let $d \in \mathbb{Z}_+$ with $d = 2^{\Omega(1/(1/2-c))}$. Consider the $(1-\alpha)$-corrupted linear regression model of Definition 1.2 with $\beta = \rho v$ for $v \in \mathcal{S}^{d-1}$, $\rho \in (0,1)$, $\sigma^2 = 1-\rho^2$. There exists an algorithm* LIST_REGRESSION_TO_TESTING *that, given a list-decoding algorithm $\mathcal{A}$ with the guarantee of returning a list $\mathcal{L}$ of candidate vectors such that for some $i \in \{1, \ldots, |\mathcal{L}|\}$, $\|\mathcal{L}(i) - \beta\|_2 \leq \rho/4$, solves the hypothesis testing Problem E.2 with probability at least $1 - |\mathcal{L}|^2 e^{-\Omega(d^{2c})}$. The running time of this reduction is quadratic in $|\mathcal{L}|$.*

*Proof.* The reduction is described in Algorithm E.2. To see correctness, first assume that the alternative hypothesis holds. We note that the rotated points $(AX_1', y_1'), \ldots, (AX_n', y_n')$ come from the Gaussian linear regression model of Definition 1.2 having $\beta' = A\beta$ as the regressor. Thus $\mathcal{A}$ finds lists $\mathcal{L}_1, \mathcal{L}_2$ such that there exist $i^* \in \{1, \ldots, |\mathcal{L}_1|\}$ with $\|\mathcal{L}_1(i^*) - \beta\|_2 \leq \rho/4$ and $j^* \in \{1, \ldots, |\mathcal{L}_2|\}$ with $\|A^T\mathcal{L}_2(j^*) - \beta\|_2 \leq \rho/4$, where we use that $A^T A = I$. Moreover, since we are considering the regression model with $\|\beta\|_2 = \rho$, $\mathcal{L}_1(i^*)$ and $A^T\mathcal{L}_2(j^*)$ must have norms belonging in $[3\rho/4, 5\rho/4]$. By the triangle inequality we get that $\|\mathcal{L}_1(i^*) - A^T\mathcal{L}_2(j^*)\|_2 \leq \rho/2$ and thus the algorithm correctly outputs $H_1$.

Now assume that the null hypothesis holds, where the marginal on points is $\mathcal{N}(0, I_d)$ and labels are independently distributed as $\mathcal{N}(0, 1/\alpha)$. Fix a pair $i \in [|\mathcal{L}_1|]$, $j \in [|\mathcal{L}_2|]$ for which $\|\mathcal{L}_1(i)\|_2, \|\mathcal{L}_2(j)\|_2 \in [3\rho/4, 5\rho/4]$. Note that, by rotation invariance of the standard Gaussian distribution and the independence between covariates and response under the null distribution, the input $\{(AX_i', y_i')\}_{i=1}^n$ for the second execution of the list-decoding algorithm is independent of $A$. Thus the list $\mathcal{L}_2$ is independent of $A$ (and also independent of $\mathcal{L}_1$). Thus, $A^T\mathcal{L}_2(j)$ is a random vector selected uniformly from the sphere of radius $\|\mathcal{L}_2(j)\|_2$ and independently of $\mathcal{L}_1(i)$. Recall that two random vectors are almost orthogonal with high probability.

---

**Algorithm 1** Reduction from Hypothesis Testing to List-Decodable Linear Regression.

---

$\mathcal{A}(\rho, (X_1, y_1), \ldots, (X_n, y_n))$: List-decoding algorithm returning a list $L$ such that $\|\mathcal{L}(i) - \beta\|_2 \leq \rho/4$ for some $i \in \{1, \ldots, |\mathcal{L}|\}$.

1: **function** LIST_REGRESSION_TO_TESTING$(\rho, (X_1, y_1), \ldots, (X_{2n}, y_{2n}))$
2:     Split dataset into two equally sized parts $\{(X_i, y_i)\}_{i=1}^{n}, \{(X_i', y_i')\}_{i=1}^{n}$.
3:     Let $A$ be a random rotation matrix independent of data.
4:     $\mathcal{L}_1 \leftarrow \mathcal{A}(\rho, (X_1, y_1), \ldots, (X_n, y_n))$.
5:     $\mathcal{L}_2 \leftarrow \mathcal{A}(\rho, (AX_1', y_1'), \ldots, (AX_{2n}', y_n'))$.
6:     **for** $i \leftarrow 1$ to $|\mathcal{L}_1|$ **do**
7:         **for** $j \leftarrow 1$ to $|\mathcal{L}_2|$ **do**
8:             **if** $\|\mathcal{L}_1(i)\|_2, \|\mathcal{L}_2(j)\|_2 \in [3\rho/4, 5\rho/4]$ and $\|\mathcal{L}_1(i) - A^T\mathcal{L}_2(j)\|_2 \leq \rho/2$ **then**
9:                 **return** $H_1$
10:     **return** $H_0$

---

**Lemma E.10** (see, e.g., [CFJ13]). *Let $\theta$ be the angle between two random unit vectors uniformly distributed over $\mathcal{S}^{d-1}$. Then we have that $\mathbf{Pr}[|cos\theta| \geq \Omega(d^{c-1/2})] \leq e^{-\Omega(d^{2c})}$ for any $0 < c < 1/2$.*

Taking a union bound over the $|\mathcal{L}_1| \cdot |\mathcal{L}_2|$ possible pairs of candidate vectors, we have that with probability at least $1 - |\mathcal{L}_1| \cdot |\mathcal{L}_2| e^{-\Omega(d^{2c})}$, for all $i \in [|\mathcal{L}_1|], j \in [|\mathcal{L}_2|]$ we have that

$$\|\mathcal{L}_1(i) - A^T\mathcal{L}_2(j)\|_2 = \sqrt{\|\mathcal{L}_1(i)\|_2^2 + \|A^T\mathcal{L}_2(j)\|_2^2 - 2(\mathcal{L}_1(i))^T(A^T\mathcal{L}_2(j))}$$

$$\geq \sqrt{2(3\rho/4)^2(1 - \Omega(d^{c-1/2}))} > \rho \, ,$$

where in the last inequality we used that $d = 2^{\Omega(1/(1/2-c))}$. This concludes correctness for the case of the null hypothesis. $\qquad\square$

We note that the Algorithm E.2 can be implemented in both of the models of computation that we consider: SQ model and low-degree polynomial test (Section F). For the SQ model, we can simulate the queries on the rotated $X$ by modifying the queries to explicitly perform the rotation on $X$ by a matrix $A$. For the low-degree polynomial test, Remark F.5 shows that this reduction can be implemented as a polynomial test.

## F  Hardness Against Low-Degree Polynomial Algorithms

In this section, we recall the recent connections between the statistical query framework and low-degree polynomials that was shown in [BBH+20], and extend our hardness results to the latter model. Sections F.1 and Section F.2 are dedicated to hypothesis problems. In Section F.3 we show that the reduction of Section E can be expressed as a low-degree polynomial test.

### F.1  Preliminaries: Low-Degree Method

We begin by recording the necessary notation, definitions, and facts. This section mostly follows [BBH+20].

**Notation**  For a distribution $D$, we denote by $D^{\otimes n}$ the joint distribution of $n$ independent samples from $D$. For $f : \mathbb{R} \to \mathbb{R}$, $g : \mathbb{R} \to \mathbb{R}$ and a distribution $D$, we define the inner product $\langle f, g \rangle_D = \mathbf{E}_{X \sim D}[f(X)g(X)]$ and the norm $\|f\|_D = \sqrt{\langle f, f \rangle_D}$. We will omit the subscripts when they are clear from the context.

**Low-Degree Polynomials**  A function $f : \mathbb{R}^a \to \mathbb{R}^b$ is a polynomial of degree at most $k$ if it can be written in the form

$$f(x) = (f_1(x), f_2(x), \ldots, f_b(x)) \, ,$$

where each $f_i : \mathbb{R}^a \to \mathbb{R}$ is a polynomial of degree at most $k$. We allow polynomials to have random coefficients as long as they are independent of the input $x$. When considering *list-decodable*

*estimation* problems, an algorithm in this model of computation is a polynomial $f : \mathbb{R}^{d_1 \times n} \to \mathbb{R}^{d_2 \times \ell}$, where $d_1$ is the dimension of each sample, $n$ is the number of samples, $d_2$ is the dimension of the output hypotheses, and $\ell$ is the number of hypotheses returned. On the other hand, [BBH$^+$20] focuses on *binary hypothesis testing* problems defined in Definition E.1.

A degree-$k$ polynomial test for Definition E.1 is a degree-$k$ polynomial $f : \mathbb{R}^{d \times n} \to \mathbb{R}$ and a threshold $t \in \mathbb{R}$. The corresponding algorithm consists of evaluating $f$ on the input $x_1, \ldots, x_n$ and returning $H_0$ if and only if $f(x_1, \ldots, x_n) > t$.

**Definition F.1** ($n$-sample $\epsilon$-good distinguisher)**.** We say that the polynomial $p : \mathbb{R}^{d \times n} \to \mathbb{R}$ is an $n$-sample $\epsilon$-distinguisher for the hypothesis testing problem in Definition E.1 if $|\mathbf{E}_{X \sim D_0^{\otimes n}}[p(X)] -$

$\mathbf{E}_{u \sim \mu} \mathbf{E}_{X \sim D_u^{\otimes n}}[p(X)]| \geq \epsilon \sqrt{\mathbf{Var}_{X \sim D_0^{\otimes n}}[p(X)]}$. We call $\epsilon$ the *advantage* of the distinguisher.

Let $\mathcal{C}$ be the linear space of polynomials with degree at most $k$. The best possible advantage is given by the *low-degree likelihood ratio*

$$\max_{\substack{p \in \mathcal{C} \\ \mathbf{E}_{X \sim D_0^{\otimes n}}[p^2(X)] \leq 1}} \left| \mathbf{E}_{u \sim \mu} \mathbf{E}_{X \sim D_u^{\otimes n}}[p(X)] - \mathbf{E}_{X \sim D_0^{\otimes n}}[p(X)] \right| = \left\| \mathbf{E}_{u \sim \mu} \left[ (\bar{D}_u^{\otimes n})^{\leq k} \right] - 1 \right\|_{D_0^{\otimes n}},$$

where we denote $\bar{D}_u = D_u/D_0$ and the notation $f^{\leq k}$ denotes the orthogonal projection of $f$ to $\mathcal{C}$.

Another notation we will use regarding a finer notion of degrees is the following: We say that the polynomial $f(x_1, \ldots, x_n) : \mathbb{R}^{d \times n} \to \mathbb{R}$ has *samplewise degree* $(r, k)$ if it is a polynomial, where each monomial uses at most $k$ different samples from $x_1, \ldots, x_n$ and uses degree at most $d$ for each of them. In analogy to what was stated for the best degree-$k$ distinguisher, the best distinguisher of samplewise degree $(r, k)$-achieves advantage $\left\| \mathbf{E}_{u \sim \mu}[(\bar{D}_u^{\otimes n})^{\leq r,k}] - 1 \right\|_{D_0^{\otimes n}}$ the notation $f^{\leq r,k}$ now means the orthogonal projection of $f$ to the space of all samplewise degree-$(r, k)$ polynomials with unit norm.

## F.2 Hardness of Hypothesis Testing Against Low-Degree Polynomials

In this section, we show the following result:

**Theorem F.2.** *Let $0 < c < 1/2$ and $m \in \mathbb{Z}_+$ with $m \leq c_1/\sqrt{\alpha}$ for some sufficiently small constant $c_1 > 0$. Consider the hypothesis testing problem of Problem E.2. For $d \in \mathbb{Z}_+$ with $d = m^{\Omega(1/c)}$, any $n \leq \Omega(d)^{(2m+1)(1/2-c)} e^{-O(m)} (1 - \rho^2)$ and any even integer $k < d^{c/4}$, we have that*

$$\left\| \mathbf{E}_{u \sim \mu} \left[ (\bar{E}_u^{\otimes n})^{\leq \infty, \Omega(k)} \right] - 1 \right\|_{R^{\otimes n}}^2 \leq 1 .$$

We prove Theorem F.2 by using the lower bound on SDA in Corollary E.8 and the relation between SDA and low-degree polynomials established in [BBH$^+$20]. In [BBH$^+$20], the following relation between SDA and low-degree likelihood ratio is established.

**Theorem F.3** (Theorem 4.1 of [BBH$^+$20])**.** *Let $\mathcal{D}$ be a hypothesis testing problem on $\mathbb{R}^d$ with respect to null hypothesis $D_0$. Let $n, k \in \mathbb{N}$ with $k$ even. Suppose that for all $0 \leq n' \leq n$, $\mathrm{SDA}(\mathcal{S}, n') \geq 100^k (n/n')^k$. Then, for all $r$, $\left\| \mathbf{E}_{u \sim \mu} \left[ (\bar{D}_u^{\otimes n})^{\leq r, \Omega(k)} \right] - 1 \right\|_{D_0^{\otimes n}}^2 \leq 1$.*

We first apply Theorem F.3 to the more general Problem E.6. In Lemma E.7 we set $n = \frac{\Omega(d)^{(m+1)(1/2-c)}}{\mathbf{E}_{y \sim R(y)}[\chi^2(A_y, \mathcal{N}(0,1))]}$ and $q = \sqrt{2^{\Omega(d^{c/2})}(n/n')}$. Then, $\mathrm{SDA}(\mathcal{S}, n') \geq \sqrt{2^{\Omega(d^{c/2})}(n/n')} \geq (100n/n')^k$ for $k < d^{c/4}$. Thus, we have shown the following.

**Corollary F.4.** *Let $0 < c < 1/2$ and the hypothesis testing problem of Problem E.6 where for every $y \in R$ the distribution $A_y$ matches the first $m$ moments with $\mathcal{N}(0, 1)$. For any $d \in \mathbb{Z}_+$ with $d = m^{\Omega(1/c)}$, any $n \leq \Omega(d)^{(m+1)(1/2-c)} / \mathbf{E}_{y \sim R(y)}[\chi^2(A_y, \mathcal{N}(0, 1))]$ and any even integer $k < d^{c/4}$, we have that*

$$\left\| \mathbf{E}_{u \sim \mu} \left[ (\bar{D}_u^{\otimes n})^{\leq \infty, \Omega(k)} \right] - 1 \right\|_{R^{\otimes n}}^2 \leq 1 .$$

*Proof of Theorem F.2.* We now apply the Corollary F.4 to Problem E.2, which is a special case of Problem E.6. The first part of Lemma 2.6 states that the distributions $A_y$'s match the first $2m$ moments with $\mathcal{N}(0,1)$ for $m \leq c_1/\sqrt{\alpha}$ and the second part implies that $\mathbf{E}_{y \sim R(y)}[\chi^2(A_y, \mathcal{N}(0,1))] = O(e^m)/(1-\rho^2)$. An application of Corollary F.4 completes the proof. $\square$

### F.3 Low-Degree Polynomial Reduction to List-Decodable Regression

**Remark F.5.** We note that the reduction of Lemma E.9 is an algorithm that can be expressed in the low-degree polynomials model. The modification of the algorithm is the following: First note that the $\ell_2$-norm of a vector is indeed a polynomial of degree two in each coordinate. Second, one can check whether there exists a pair $i \in [|\mathcal{L}_1|], j \in [|\mathcal{L}_2|]$ with $\|\mathcal{L}_1(i)\|_2, \|\mathcal{L}_2(j)\|_2 \in [3\rho/4, 5\rho/4]$ for which $\|\mathcal{L}_1(i) - A^T \mathcal{L}_2(j)\|_2 \leq \rho/2$ using the condition

$$\sum_{i \in 1}^{|\mathcal{L}_1|} \sum_{j \in 1}^{|\mathcal{L}_2|} \mathbf{1}(\|\mathcal{L}_1(i)\|_2^2 \geq (3\rho/4)^2) \cdot \mathbf{1}(\|A^T \mathcal{L}_2(j)\|_2^2 \leq (5\rho/4)^2) \cdot \mathbf{1}(\|\mathcal{L}_1(i) - A^T \mathcal{L}_2(j)\|_2^2 \leq \rho^2/4) = 0 \;,$$

and use a polynomial approximation for the step function in order to express each term as a polynomial. The degree needed for a uniform $\epsilon$-approximation has been well-studied [GR08, Gan02, EY07].

**Lemma F.6** ([EY07]). *Let $f : \mathbb{R} \to \mathbb{R}$ be the step function defined as $f(x) = 1$ for all $x \geq 0$ and $f(x) = 0$ otherwise. The minimum $k \in \mathbb{Z}_+$ for which there exists a degree-$k$ polynomial $p : \mathbb{R} \to \mathbb{R}$ such that $\max_{x \in [-1,1]} |f(x) - p(x)| \leq \epsilon$ is $k = \Theta(1/\epsilon^2)$.*

For our purpose, it suffices to approximate the step function up to error $\epsilon = \Theta(1/(|\mathcal{L}_1| \cdot |\mathcal{L}_2|))$, thus the resulting polynomial test has degree $\Theta(|\mathcal{L}_1|^2 \cdot |\mathcal{L}_2|^2)$.

## G Additional Technical Facts

Our bounds in Lemma 2.6 required the standard fact below. Here we provide its proof for completeness.

**Fact G.1.** *For any one-dimensional distribution $P$ that matches the first $m$ moments with $\mathcal{N}(0,1)$ and has $\chi^2(P, \mathcal{N}(0,1)) < \infty$ the following identity is true*

$$\chi^2(P, \mathcal{N}(0,1)) = \sum_{i=m+1}^{\infty} \left( \mathop{\mathbf{E}}_{X \sim P}[h_i(X)] \right)^2 \;.$$

*Proof.* . Let $\phi$ denote the pdf of the standard one-dimensional Gaussian. For this proof, we use a slightly different definition of the space $L^2(\mathbb{R}, \mathcal{N}(0,1))$. We define it as the space of functions for which $\int_{\mathbb{R}} f^2(x)/\phi(x)\mathrm{d}x < \infty$ with the inner product $\langle f, g \rangle := \int_{\mathbb{R}} f(x)g(x)/\phi(x)\mathrm{d}x$ (note the similarity with the definition of $\chi^2$-divergence). The *Hermite functions* (or often called *Hermite-Gauss functions*) $h_i(x)\phi(x)$ for $i = 0, 1, \ldots$ form a complete orthonormal basis of the space $L^2(\mathbb{R}, \mathcal{N}(0,1))$ with respect to that inner product. It is easy to check that this statement is equivalent to the statement that Hermite polynomials $\{h_i\}_{\mathbb{N}}$ form a complete orthonormal basis of the space of all functions $f : \mathbb{R} \to \mathbb{R}$ for which $\mathbf{E}_{x \sim \mathcal{N}(0,1)}[f^2(x)] < \infty$ (i.e., our old definition of $L^2(\mathbb{R}, \mathcal{N}(0,1))$). . Since $\chi^2(P, \mathcal{N}(0,1)) < \infty$ we have $P \in L^2(\mathbb{R}, \mathcal{N}(0,1))$ and thus we can write $P(x) = \sum_{i=0}^{\infty} a_i h_i(x)\phi(x)$, where $a_i = \mathbf{E}_{X \sim P}[h_i(X)]$. Using the fact that $P$ agrees with the first $m$ moments of $\mathcal{N}(0,1)$ and the property of Hermite polynomials $\mathbf{E}_{X \sim \mathcal{N}(0,1)}[h_i(X)] = \mathbf{1}(i = 0)$ we get that $a_0 = \mathbf{E}_{X \sim \mathcal{N}(0,1)}[h_0(X)] = 1$ and $a_i = \mathbf{E}_{X \sim \mathcal{N}(0,1)}[h_i(X)] = 0$ for $0 < i \leq m$. Thus

$$P(x) = \phi(x) + \sum_{i=m+1}^{\infty} a_i h_i(x)\phi(x) \;.$$

The $\chi^2$-divergence can then be written as

$$\chi^2(P, \mathcal{N}(0,1)) = \int_{\mathbb{R}} \frac{(P(x) - \phi(x))^2}{\phi(x)}\mathrm{d}x = \int_{\mathbb{R}} \frac{1}{\phi(x)} \left( \sum_{i=m+1}^{\infty} a_i h_i(x)\phi(x) \right)^2 \mathrm{d}x = \sum_{i=m+1}^{\infty} a_i^2 \;,$$

where the last part uses orthonormality of the functions $h_i(x)\phi(x)$. $\square$

We now turn to Claim B.3 which is restated below.

**Claim G.2.** *If $P = \sum_{i=1}^{k} \lambda_i N(\mu_i, \sigma_i^2)$ with $\mu_i \in \mathbb{R}$, $\sigma_i < \sqrt{2}$ and $\lambda_i \geq 0$ such that $\sum_{i=1}^{k} \lambda_i = 1$, we have that $\chi^2(P, \mathcal{N}(0,1)) < \infty$.*

For that we need the following two facts about $\chi^2$-distance between Gaussians. Their proofs can be done by direct calculations.

**Fact G.3.** *Let $k \in \mathbb{Z}_+$, distributions $P_i$ and $\lambda_i \geq 0$, for $i \in [k]$ such that $\sum_{i=1}^{k} \lambda_i = 1$. We have that $\chi^2\left(\sum_{i=1}^{k} \lambda_i P_i, D\right) = \sum_{i=1}^{k} \sum_{j=1}^{k} \lambda_i \lambda_i \chi_D(P_i, P_j)$.*

*Proof.*

$$\chi^2\left(\sum_{i=1}^{k} \lambda_i P_i, D\right) + 1 = \int_{\mathbb{R}} \left(\sum_{i=1}^{k} \lambda_i P_i(x)\right)^2 / D(x)\mathrm{dx} = \sum_{i=1}^{k} \sum_{j=1}^{k} \lambda_i \lambda_j \int_{\mathbb{R}} P_i(x) P_j(x) / D(x)\mathrm{dx}$$

$$= \sum_{i=1}^{k} \sum_{j=1}^{k} \lambda_i \lambda_j \left(\chi_D(P_i, P_j) + 1\right) = \sum_{i=1}^{k} \sum_{j=1}^{k} \lambda_i \lambda_j \chi_D(P_i, P_j) + \left(\sum_{i=1}^{k} \lambda_i\right)^2$$

$$= \sum_{i=1}^{k} \sum_{j=1}^{k} \lambda_i \lambda_j \chi_D(P_i, P_j) + 1 \ .$$

$\square$

**Fact G.4.**

$$\chi_{\mathcal{N}(0,1)}\left(\mathcal{N}(\mu_1, \sigma_1^2), \mathcal{N}(\mu_2, \sigma_2^2)\right) = \frac{\exp\left(-\frac{\mu_1^2(\sigma_2^2 - 1) + 2\mu_1\mu_2 + \mu_2^2(\sigma_1^2 - 1)}{2\sigma_1^2(\sigma_2^2 - 1) - 2\sigma_2^2}\right)}{\sqrt{\sigma_1^2 + \sigma_2^2 - \sigma_1^2\sigma_2^2}} - 1 \ .$$

The proof of Claim G.2 then consists of applying Fact G.3 and using Fact G.4 for each one of the generated terms.