# OpenReview forum: "Statistical Query Lower Bounds for List-Decodable Linear Regression"
_NeurIPS.cc/2021/Conference — NeurIPS 2021 Spotlight_

### Official Review · Reviewer_A5qv · 2021-07-12

**Rating:** 7
**Confidence:** 4

**Summary:**

The paper the list-decodable linear regression problem, where the adversary can corrupt a fraction alpha>0.5 of the labels, and the goal is to return a list of vectors of size f(alpha) such that at least one of the vectors is close to the true regression vector. The paper proves computational-statistical tradeoffs for the problem in the statistical query (SQ) model. It shows that any SQ algorithm needs d^(poly(1/alpha)) samples to solve the problem, whereas d/alpha^3 samples are information theoretically sufficient. Therefore, the previous known algorithms for the problem are nearly optimal in the SQ model.

To prove this result, the paper leverages and builds on the machinery in DKS17. DKS17 show that it is possible to prove SQ lower bounds by showing that there exists a family of distributions whose projection onto a certain direction v matches a given number of moments of the standard Gaussian, and the distribution is a Gaussian orthogonal to v. The paper shows that it is possible to match poly(1/alpha) moments onto this direction v, which gives a d^poly(1/alpha) lower bounds using the framework of DKS17. To match these moments, the paper employs an innovative non-constructive method which uses a relation between moments of a distribution and non-negative polynomials. Several technical challenges are involved in this construction, these are briefly outlined in Section 1.3.

**Limitations And Societal Impact:**

Yes, limitations and potential negative societal impact are adequately addressed.

**Main Review:**

The paper is technically interesting, and employs an interesting suite of techniques to obtain the desired SQ lower bound. It is not immediately clear though if these techniques can be useful in other contexts, and at least some of the heavy lifting here is done in DKS17 and DKS19. That being said, it still appears to be a solid technical result which obtains matching lower bounds in the SQ model.

I'm not convinced though that the conceptual message is so strong here. A d^2 lower bound---and probably a d^constant lower bound as well---follows from DKS17. Though the authors make some references to crowd sourcing, it is not clear if the asymptotic dependence on alpha is that important from a practical perspective at least. From the conceptual side, the authors point that the result implies that list-decodable linear regression is harder than the mixture of linear-regressions, but its not clear why the reader should find this surprising or take away from it. Maybe elaborating more on what the SQ lower bound construction achieves which moves the problem away from a mixture of linear regressions could help here. Due to all this, the conceptual message appears to be a bit limited here.

The paper is overall well-written. However, I suggest that it would be better to spend much more time on the outline in Section 1.3, and move some of the detailed proofs in the main body to the appendix. As it is, it is not easy to follow Section 1.3 without first understanding some of the later sections in at least some depth.

---post author response---

I thank the authors for the very detailed and concrete response to my questions and concerns. I am satisfied regarding the conceptual contribution here, and have therefore raised my score.

**Time Spent Reviewing:**

3

---

> ### Author Response · Authors · 2021-08-10
> **Author Response**
>
> We thank the reviewer for their time and effort in evaluating our work, and finding it interesting, innovative, and strong in technical aspects. We address their comments below:
>
> 1. The reviewer states: *"A $d^2$ lower bound---and probably a $d^\text{constant}$ lower bound as well---follows from DKS17. ….., it is not clear if the asymptotic dependence on $\alpha$ is that important from a practical perspective”*
>
>     We disagree with the reviewer here on three points: (i) significant difference between a $d^2$ and a $d^{\textrm{poly}(1/\alpha)}$ SQ lower bound, (ii) applicability of a $d^2$ SQ lower bound from prior work, and (iii) practical implications of our lower bound. We elaborate on each point below.
>
>    +  (i)  The parameter $\alpha$, quantifying the fraction of inliers, is part of the input to the list-decodable learning problem, and can be arbitrarily small. The two stated bounds  ($d^2$ versus $d^{\textrm{poly}(1/\alpha)}$) are *super-polynomially* related. Our work shows that $\textrm{poly}(d/\alpha)$ samples suffice to solve the problem; while any sub-exponential time SQ algorithm requires at least $d^{\textrm{poly}(1/\alpha)}$ samples. We view this statement as conceptually interesting and arguably surprising. Also see response to point (iii) below regarding the practical relevance of our lower bound.
>
>    + (ii) We are a bit puzzled by exactly which prior SQ lower bound the reviewer is referring to. Previous SQ lower bounds for similar problems have studied the small noise regime (minority of outliers) and required that the algorithm outputs a single very accurate hypothesis. Our current lower bound says that in the large error regime (majority of outliers) we cannot obtain a *very weak* approximation to the regressor, even if we are allowed to output a sub-exponential sized list of hypotheses. If the reviewer is referring to the SQ lower bound for linear regression in [DKS19], we note that this lower bound crucially relies on the covariance matrix of the clean data being unknown; while in our current setting the clean data is assumed to have identity covariance. (And, even then,  [DKS19] establishes that we cannot approximate the regressor to within error better than $\epsilon^{1/2}$, where $\epsilon < 0.5$ is the fraction of outliers in that work.) In summary, this prior SQ lower bound does not have any direct implications to the list-decodable setting.
>
>     +  (iii) Practical implications: As mentioned in the introduction of our submission, in several
> practical settings, the fraction of inliers $\alpha$ is very small (goes to $0$). See, for example, the cited works of [MV18, CSV17]. The asymptotic dependence on $\alpha$ is of crucial importance in such settings.
>
>
> 2. Regarding the reviewer's comment *"From the conceptual side, the authors point that the result implies that list-decodable linear regression is harder than the mixture of linear-regressions, but its not clear why the reader should find this surprising or take away from it"*
>
>     We outline some conceptually interesting implications of our results below:
>
>    + (i) We note that the problems of list-decoding the mean of a Gaussian, learning a Gaussian Mixture Model (GMM) with spherical components, and learning a Mixture of Linear Regressions (MLR) all have quasipolynomial time algorithms, as a function of $1/\alpha$, with essentially optimal error [DKS18, DK20]. It is therefore somewhat surprising that the problem of list-decodable linear regression does not (in the SQ model).
>
>    +  (ii) Although list-decodable mean estimation is more general than learning spherical GMMs, the best-known results for such GMMs can be achieved by reducing it to list-decodable mean estimation (see [DKS18]). Our results imply that such an algorithmic result is not possible for the setting of linear-regression: there is a quasipolynomial time algorithm for MLR [DK20], but our lower bounds show an exponential dependence in $\textrm{poly}(1/\alpha)$ for the list-decodable version.
>
>    +   (iii) Overall, our results show that list-decodable linear regression behaves very differently from list-decodable mean estimation. This different behavior is specific to the list-decodable setting (majority of outliers), and is not observed in the traditional robust statistics (minority of outliers).

---

### Official Review · Reviewer_8XsW · 2021-07-13

**Rating:** 8
**Confidence:** 4

**Summary:**

This paper proves a super-polynomial statistical query lower bound for the problem of of list-decodable linear regression where an adversary can corrupt a majority of the examples. Specifically, the authors show a $d^{\text{poly}(1/\alpha)}$ lower bound, which matches the state-of-the-art algorithmic upper bounds; this gives an evidence that the gap between the statistical and computational gap is inherent.



**Ethical Concerns:**

No.

**Limitations And Societal Impact:**

Yes.

**Main Review:**

The paper is well-written, motivated, rigorous, and interesting. Despite the fact that it is quite a technical paper, the authors managed to present their ideas and results in a clear and neat way. I read most of the proofs and everything seems clean to me.
I do not have any important comments/concerns, and therefore I am happy to recommend acceptance of this paper.


**Time Spent Reviewing:**

72

---

> ### Author Response · Authors · 2021-08-10
> **Author Response**
>
> We thank the reviewer for their time, effort, and positive assessment of our work.

---

### Official Review · Reviewer_9D9p · 2021-07-15

**Rating:** 8
**Confidence:** 4

**Summary:**

This paper studies linear regression in the "list-decodable learning" setting.
The main result is a computational lower bound shows that existing algorithms are qualitatively optimal, at least among statistical query algorithms.

**Limitations And Societal Impact:**

This is a theory paper with limited or no direct societal impact.

**Main Review:**

This paper studies linear regression in the "list-decodable learning" setting.
Concretely, the problem studied is as follows.
One receives samples $(X_1,y_1),\ldots,(X_n,y_n)$ in $R^{d+1}$ with the following guarantee.
For some $\alpha > 0$, some $\alpha$-fraction of the samples was drawn according to the standard linear model: $y_i = \beta^\top X_i + \eta_i$ where $\eta_i$ is independent Gaussian noise and $\beta$ is an unkonwn regression vector.
The remaining samples may be arbitrary.

Crucially, $\alpha$ may be very small -- a large majority of the samples can be corrupted.
While it is information-theoretically impossible to identify $\beta$ from such highly corrupted samples, one can actually find a list of $O(1/\alpha)$ candidates $\beta'$ such that there exists $\beta'$ in the list such that $\|\beta - \beta'\|$ is small.

List-decodable learning in general (for clustering, linear regression, community detection, and other problems) has been intensely studied in recent years, as it is an appealing common generalization of learning in mixture models and robust statistics. (And has additional applications, e.g. to learning with a few verified samples.)
Notably, in many settings the complexity of list-decodable learning is no greater than the complexity of learning mixture models in the setting.

The main result in this paper is a computational lower bound for list-decodable linear regression.
Existing efficient algorithms which find a list of size $\poly(1/\alpha)$ require $d^{\poly(1/\alpha)}$ samples and time.
(Even though, as this paper shows, in exponential time $\poly(d/\alpha)$ samples suffice.)
This paper shows that this is in fact necessary for statistical query algorithms.
This is somewhat surprising, because the analogous mixtures of linear regression problem can be solved in quasipolynomial time and samples (roughly, $(d/\alpha)^{\poly \log (d/\alpha)}$, which is almost exponentially faster.

The statistical query lower bound also implies lower bounds for algorithms based on low-degree polynomials, via a known reduction.

The lower bound requires constructing a hard family of instances.
The construction in this paper builds on a framework established by Diakonikolas-Kane-Stewart '17, which has since been used to prove numerous SQ lower bounds for high-dimensional statistics.
The construction involves finding a family of distributions which fit the list-decodable linear regression model but whose moments match those of a Gaussian.
The main innovation here over prior applications of [DKS17] technique is that the distributions need to match a growing number of moments -- $\poly(1/\alpha)$.
The authors use LP duality to show the existence of such distributions, rather than giving an explicit construction.

By this point there are numerous SQ lower bounds for high-dimensional (robust) learning.
This paper stands out from the crowd in the following ways:
-- it qualitatively settles the complexity of list-decodable regression; this was previously unstudied from a lower bounds perspective
-- it combines LP duality with the DKS17 technique
-- it separates the complexity of a natural list-decodable learning problem from the analogous mixture model problem.

I recommend that the paper be accepted, possibly as a spotlight.

(I did not carefully check mathematical details, but the construction seems reasonable.)

**Time Spent Reviewing:**

3

---

> ### Author Response · Authors · 2021-08-10
> **Author Response**
>
> We thank the reviewer for their time, effort, and positive assessment of our work.

---

### Official Review · Reviewer_9Tob · 2021-07-27

**Rating:** 8
**Confidence:** 4

**Summary:**

Consider the following 'robust' linear regression problem: We see samples of the form $(X,Y)$ where with probability $\alpha$ the sample is drawn from a true linear-regression problem with Gaussian covariates (i.e., $Y = \beta^T X + Noise$) and an arbitrary noise distribution with probability $1-\alpha$.

In the above scenario, it is impossible to identify the unknown regressor vector $\beta$ exactly. However, one can ask for 'list-decodable linear regression'. Several recent works studied the case where we want to identify a small set of candidate vectors (preferably of size $(1/\alpha)^{O(1)}$) that will contain the true vector $\beta$ (or something close to it).

Two recent works showed positive results in this direction obtaining $\approx poly(1/\alpha)$ list sizes but the run-time was $d^{(1/\alpha)^{O(1)}}$.

The present paper shows that such a dependence on $\alpha$ is in some sense necessary: For the widely studied class of SQ algorithms, one needs an oracle with tolerance at most $\approx d^{-1/\sqrt{\alpha}}$. Thus suggesting that the above works are almost optimal.

List-decodable linear regression is an important problem from a theoretical perspective as it is connected to many other fundamental learning problems (e.g., learning mixture models). The positive results on the problem used the powerful SoS hierarchy and while the run-time was polynomial, it wasn't really a nice polynomial. The present paper helps us partly understand why this is the case and gives some evidence that beating these would be unlikely.

The methods of the paper are quite nice and will likely find use elsewhere as well. I recommend acceptance.

**Ethical Concerns:**

None.

**Limitations And Societal Impact:**

Yes.

**Main Review:**

Techniques: The paper builds on the main way of showing SQ lowerbounds by designing distributions that satisfy the divergence conditions as in [FGR+]. This in turn relies heavily on the conceptual idea of [DKS] who suggested coming up with distributions that look Gaussian in all but one direction and then moment match a Gaussian in one hidden direction. In most cases, if one matches $k$ moments, one gets a lower bound of roughly $d^{\Omega(k)}$. Implementing this conceptual idea for the present scenario is not easy and the authors develop several clever ideas. For instance, one such idea is that they generate the noise distribution such that the moment matching (using noise) happens on conditional distributions $X | Y= y_0$ for various $y_0$. The authors have to cleverly trade-off the noisy fraction for larger $y_0$ so that the overall noise is small while still matching moments for large $y_0$.

Couple of questions:
1. To my understanding, the SQ lower bound (and the implied lower bound to low-degree algorithms) does not apply to the SoS algorithms of [KKK] and [RY]. While this is not a big deterrent against the present lower bound, it would be good to mention that this is the case (if it is so).
2. For intuition, to help the readers understand the construction it might be good to say a sentence or two as to why one cannot threshold on the value of $y$ and hence work with 'better' noise rates on the small $y$ and hence identify the $\beta$ better.

**Time Spent Reviewing:**

3

---

> ### Author Response · Authors · 2021-08-10
> **Author Response**
>
> We thank the reviewer for their time, effort, and positive assessment of our work. We address their specific questions below:
>
> 1. *Sum-of-squares based algorithms vs SQ lower bound*
>
>    +  Yes, the reviewer is correct in their understanding. However, we believe the particular SoS algorithms of [KKK19, RY20] for the problem we study can be simulated in the SQ framework with qualitatively similar performance guarantees.
>     + In general, SQ lower bounds do not imply lower bounds against all SoS algorithms, and SoS lower bounds do not imply SQ lower bounds. At the same time, lower bounds against low-degree tests have become the standard heuristic for SoS lower bounds, and we establish hardness against low-degree tests as well. Moreover, our results (hardness against SQ algorithms and low-degree tests) are the only known evidence of hardness for this problem.
>
> 2. *Thresholding the value of $y$*
>
>    We are unclear as to what the reviewer means by this question. In our SQ lower bound, we show SQ-hardness against the Gaussian distribution, which has unbounded support. If we thresholded the value of $y$ and truncated the distribution, then we would be dealing with a stronger contamination model that can also remove a fraction of the clean data.

---

### Author Response · Authors · 2021-08-10
**Joint Response**

We thank the reviewers for their time and effort in providing feedback. We are encouraged by the universally positive scores, and that all the reviewers appreciated the paper for the following: (i) *significant results* (9Tob, 9D9p, 8XsW), (ii) *technical contribution* (9Tob, 9D9P, 8XsW, A5qv), and (iii) *clarity* (8XsW, A5qv). We address the individual questions and comments by the reviewers separately.

---

### Decision · Program_Chairs · 2021-09-28

**Decision:**

Accept (Spotlight)

**Comment:**

This paper proves an SQ lower bound for the problem of list-decodable linear regression over gaussian covariates that suggests that the algorithm needs a running time of d^{poly(1/\alpha)} to solve the problem in d dimensions with \alpha fraction inliers. This matches the performance of the best-known algorithm for the problem and shows that list-decodable linear regression is likely harder than the related special case of mixed linear regression that admits a sub-exponential time algorithm (in 1/\alpha).

In authors' ​response, it was clarified that while SQ lower bound don't apply to SoS based algorithms, the authors believe that it does apply to the specific algorithm employed in prior work on this problem. This is a somewhat non-trivial statement and was brought up in reviewer discussion -- we recommend adding formal proof of this claim to the camera-ready version.

The authors' response also clarified that prior works on SQ lower bound only yield a fixed polynomial in d lower bound on the running time resolving the concern about the relationship to prior works on SQ lower bounds for related problems.

The paper proves an interesting lower bound on an important problem in algorithmic robust statistics and shows an interesting "separation" between list-decodable learning and the problem of learning components of a mixture. The techniques build on and improve on the SQ lower bounds proven in prior works. We recommend acceptance.

**Consistency Experiment:**

NeurIPS has a long history of experimentation. In 2014, NeurIPS ran an experiment in which 10% of submissions were reviewed by two independent committees to quantify the randomness in the review process. This year, we repeated a variant of this experiment to see how the quality of the review process has changed over time.  This paper was part of the experiment and was therefore assigned to two committees (consisting of reviewers, an Area Chair, and a Senior Area Chair) that reached independent decisions.  If both committees made the same recommendation, this recommendation was followed. If a single committee recommended acceptance, the paper was accepted (with the exception of a few cases in which the other committee identified what we considered a fatal flaw, e.g., an error in a key result).

Both committees reached the same decision: **Accept (Spotlight)**

The other committee assigned to the paper recommended **Accept (Spotlight)**.  You can find the other set of reviews, along with any follow up discussion with the authors here:
https://openreview.net/forum?id=neRSyESg1GU